EMBO
Molecular Medicine

# Disruption of Sema3A/Plexin-A1 inhibitory signalling in oligodendrocytes as a therapeutic strategy to promote remyelination

Fabien Binamé[1,†], Lucas D Pham-Van[1,†], Caroline Spenlé[1], Valérie Jolivel[1], Dafni Birmpili[1], Lionel A Meyer[1], Laurent Jacob[1], Laurence Meyer[1], Ayikoé G Mensah-Nyagan[1], Chrystelle Po[2], Michaël Van der Heyden[1], Guy Roussel[1] & Dominique Bagnard[1,*] iD

## Abstract

Current treatments in multiple sclerosis (MS) are modulating the inflammatory component of the disease, but no drugs are currently available to repair lesions. Our study identifies in MS patients the overexpression of Plexin-A1, the signalling receptor of the oligodendrocyte inhibitor Semaphorin 3A. Using a novel type of peptidic antagonist, we showed the possibility to counteract the Sema3A inhibitory effect on oligodendrocyte migration and differentiation *in vitro* when antagonizing Plexin-A1. The use of this compound *in vivo* demonstrated a myelin protective effect as shown with DTI-MRI and confirmed at the histological level in the mouse cuprizone model of induced demyelination/remyelination. This effect correlated with locomotor performances fully preserved in chronically treated animals. The administration of the peptide also showed protective effects, leading to a reduced severity of demyelination in the context of experimental autoimmune encephalitis (EAE). Hence, the disruption of the inhibitory microenvironmental molecular barriers allows normal myelinating cells to exert their spontaneous remyelinating capacity. This opens unprecedented therapeutic opportunity for patients suffering a disease for which no curative options are yet available.

**Keywords** multiple sclerosis; myelination; oligodendrocyte; Plexin; Semaphorin
**Subject Categories** Pharmacology & Drug Discovery; Neuroscience

## Introduction

The efficient propagation of electrical nerve impulses requires the establishment of myelin sheaths around axons (Baumann & Pham-Dinh, 2001). Multiple sclerosis (MS) is a disease affecting nerve conduction as a consequence of myelin damages. Current treatments of MS are not curative but only fighting the associated inflammation without repairing myelin. The myelination process is orchestrated by integrated cellular and molecular interactions unravelling potential therapeutic strategies for remyelination (Mi *et al*, 2007; Abu-Rub & Miller, 2018). During development, precursors of oligodendrocytes (OL), the myelinating cells of the central nervous system (CNS), use guidance cues to migrate towards their target neuronal cells. Strikingly, this process is not restricted to developmental phases but also occurs to a certain extent in the adult brain. This is also the case in CNS-demyelinating diseases in which partial spontaneous remyelination closely mimicking developmental myelination is observed. Combinations of soluble factors and membrane-bound cues are considered to create a specific molecular environment dictating the precise recognition process leading to the myelination of axons (Piaton *et al*, 2010). Among these factors regulating the early phase of myelination, members of the Semaphorin family have been shown to regulate OL precursor cell migration in the optic nerve (Tsai & Miller, 2002) and inhibit adult OL process outgrowth (Ricard *et al*, 2001). Strikingly, an analysis of human multiple sclerosis sample tissues and the use of an experimental model of demyelination revealed a clear spatio-temporal regulation of Sema3A expression, thereby strengthening the idea of a role of Semaphorins in myelination (Williams *et al*, 2007). However, little is known about the expression of Semaphorin receptors in this context. This is indeed a crucial issue because the biological functions of Semaphorins are intimately linked to the composition of a receptor complex associating various partners in order to trigger appropriate growth-promoting or growth-inhibiting effects of Semaphorins (Derijck *et al*, 2010). Thus, we decided to investigate the expression of Plexin-A1, one of the major Sema3A-transducing receptors so far essentially described as a transducer of Sema3A inhibiting signal in neurons (Püschel, 2002) or for its role in the immune system (O'Connor & Ting, 2008). We found that Plexin-A1 is overexpressed in the white matter of MS patients. Moreover, in

1   INSERM U1119 Biopathology of Myelin, Neuroprotection, Therapeutic Strategy, Labex Medalis, Fédération de Médecine Translationnelle de Strasbourg, Strasbourg University, Strasbourg, France
2   Institut de Physique Biologique, Faculté de Médecine, Strasbourg University, Strasbourg, France
    *Corresponding author. Tel: +33-3-68-85-71-50; E-mail: bagnard@unistra.fr
    †These authors contributed equally to this work

mouse, only few CNP-positive OL expressed Plexin-A1. However, we observed a sevenfold increase in CNP-positive OL in animals exhibiting cuprizone-induced lesions. This was concomitant to local deposition of Sema3A in the demyelinated regions. Moreover, *in vitro* studies showed that blocking Plexin-A1 counteracted the anti-migratory and anti-differentiation effect of Sema3A in oligodendrocytes. Hence, we showed that the administration of the Plexin-A1 antagonist peptide MTP-PlexA1 improved myelin content and locomotor activity in mice fed with cuprizone to induce demyelinated lesions or in the context of experimental autoimmune encephalomyelitis (EAE). Altogether, our results suggest a therapeutic potential of inhibiting Plexin-A1 in myelin diseases such as multiple sclerosis in which we found overexpression of Plexin-A1.

# Results

## Plexin-A1 is expressed in human oligodendrocytes

Previous studies showed the expression of Sema3A in MS lesions (Williams *et al*, 2007) where it is supposed to contribute to the lack of remyelination. We now examined whether OL express the Sema3A signalling receptor Plexin-A1. To this end, we performed immunocytochemical staining for Plexin-A1 on a human brain tissue array. The results confirm previous data (Jacob *et al*, 2016), showing neuronal expression of Plexin-A1 in several CNS locations, but also demonstrate the expression in oligodendrocyte cells in the white matter (Fig 1A). The global analysis of the tissue array revealed a wide expression in several regions of the central nervous system including the cortex, striatum or the spinal cord (data not shown). To confirm the identity of Plexin-A1-expressing cells in the white matter, we performed a CNP (3′,5′-cyclic nucleotide phosphodiesterase) staining (pan-marker of oligodendrocytes) on the adjacent section of the tissue array. False colour coding of the microphotographs allowed overlay of the two sections to exemplify the co-expression of Plexin-A1 and CNP (Fig 1B).

## Plexin-A1 is overexpressed in MS patients

In order to evaluate the Plexin-A1 level of expression in the context of multiple sclerosis, we first performed data mining from published gene array profiles using the GEO platform. The analysis of four chronic plaques and two healthy controls (Han *et al*, 2012) revealed an averaged 4.2-fold increase in *Plexin-A1* mean expression (4.7-fold increase in the median) and 3.2-fold increase in *SEMA3A* mean expression (2.8-fold increase in the median) in the disease condition (Appendix Fig S1). We next collected and analysed white matter samples of 11 MS patients and nine healthy controls from the Netherlands Brain Bank (see Appendix Table S1 for details). We performed a Western blot analysis to evaluate Plexin-A1 content and found a 2.3-fold increased expression in MS patients (Fig 2A and B). The proportion of MS patients exhibiting such a twofold increase in Plexin-A1 expression above the averaged expression measured in healthy controls reached 45% of the patients (Fig 2C). To further characterize this overexpression of Plexin-A1, we also determined the number of CNP-Plexin-A1-positive cells by immunocytochemistry conducted on fresh-frozen sections of the white matter samples. As seen in Fig 2D and E, we found a threefold

increase in Plexin-A1-positive CNP-expressing cells in MS patients compared to healthy controls. This suggested that Plexin-A1 may represent an interesting target in the context of MS.

## Plexin-A1 is overexpressed in OL in experimental demyelination conditions

To address whether Plexin-A1 may be involved in demyelination/remyelination conditions in the adult, we used the cuprizone model. In this model, a demyelination/remyelination process is obtained by feeding mice with 0.3% cuprizone (*bis*-cyclohexanone oxal-dihydrazone) for 4 weeks (acute demyelination phase) and normal diet for 2 weeks (initiation of remyelination) or 4 weeks (induction of total remyelination; see Fig 3A for representative examples). Administration of 8-week cuprizone diet induces permanent demyelination as described previously (Matsushima & Morell, 2001). We performed double immunostaining for CNP and Plexin-A1 and determined at the corpus callosum level the number of double-positive cells in the different demyelination/remyelination status (Fig 3B). Only few OL expressed Plexin-A1 in control conditions (normal diet, 5.3% of CNP-positive OL). However, we found that Plexin-A1 was significantly overexpressed in OL after 4-week cuprizone and 2-week normal diet administration (37.8% of CNP-positive OL, ANOVA, $P = 0.0015$; Fig 3C). Strikingly, Plexin-A1 expression was back to control level after total recovery (4-week cuprizone + 4-week normal diet), while it was not overexpressed in the 8-week cuprizone group (Fig 3C). Moreover, similar to what has been previously described in human samples of multiple sclerosis (Williams *et al*, 2007), we also found that Sema3A expression transiently increased in acute phases of cuprizone-induced demyelination (Fig 3D). The overexpression of Sema3A was not uniform throughout the brain but rather matched with cuprizone-induced lesion sites. This suggested that a Sema3A/Plexin-A1 signalling is reactivated in the adult in case of demyelination/remyelination process.

## MTP-PlexA1 cancels Sema3A repulsive effect on oligodendrocyte migration

Sema3A deposit inhibits OPC recruitment into MS lesions, explained in part by Sema3A repulsive effect (Boyd *et al*, 2013). We first addressed the involvement of Plexin-A1 in Sema3A signalling by RNA interference. We obtained a significant 50% knockdown of Plexin-A1 in Oli-neu cells as seen by immunocytochemistry and RT–qPCR (Fig 4A). We used XCELLigence transwell chambers to monitor 2% serum-induced chemotactic migration of the OPC cell line Oli-neu during 8 h. Compared to control migration without Sema3A, addition of 20 ng/ml Sema3A in the lower chamber decreased migration of 37% of Oli-neu cells transfected with siRNA control. Oli-neu cells transfected with siRNA targeting Plexin-A1 reached 95% of control migration (Fig 4B). This result confirmed the requirement of Plexin-A1 to drive the inhibitory effect of Sema3A. This lower expression was indeed sufficient to significantly decrease the number of NRP1/Plexin-A1 dimers (a key step to trigger semaphoring signalling) as determined by proximity ligation assay (Fig 4C). We next used the recently developed peptidic antagonist MTP-PlexA1. This peptide blocks receptor dimerization and signalling by interfering with the transmembrane domain of Plexin-A1. It has been successfully used *in vitro* to antagonize Plexin-A1

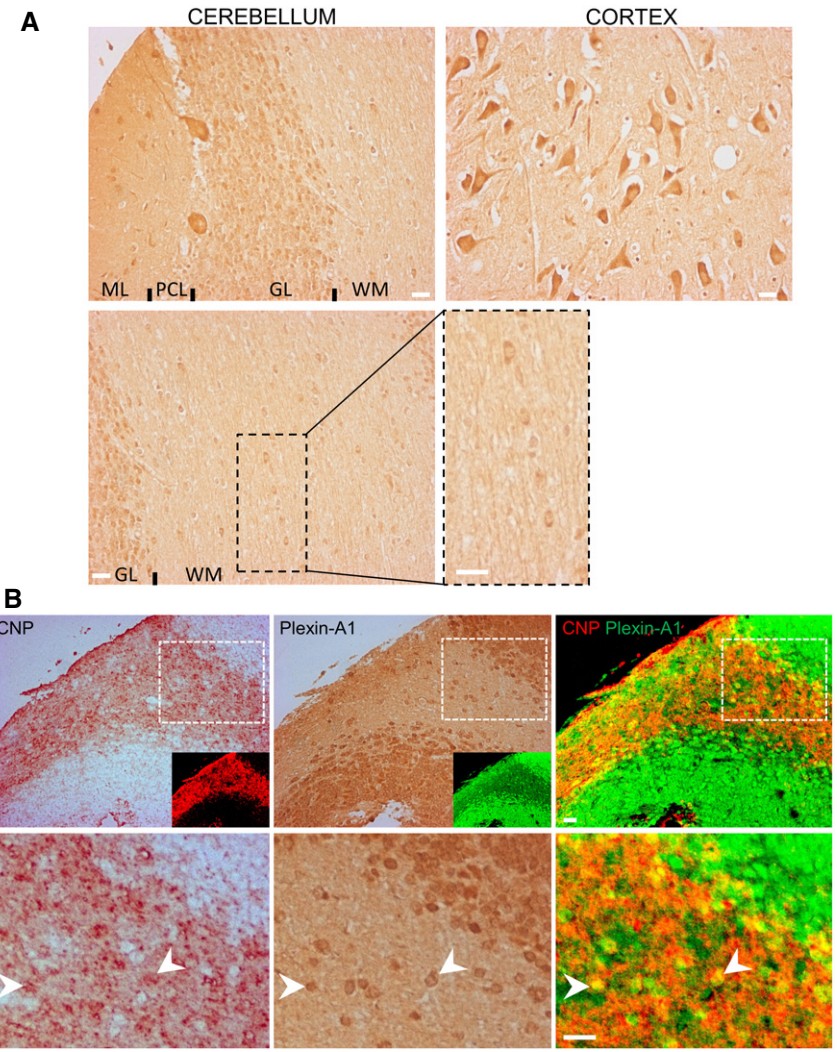

**Figure 1. Expression of Plexin-A1 in the human brain.**

A  Microphotographs showing the expression of Plexin-A1 on a section of human brain array in the cerebellum (ML: molecular layer; PC: Purkinje cell layer; GL: granular layer; WM: white matter) and in the cortex. Oligodendrocytes in the white matter are expressing Plexin-A1. Scale bar = 10 μm.

B  Immunostainings of Plexin-A1 and CNP (pan-oligodendrocyte marker) were conducted on adjacent sections of a human brain tissue array. Overlay of adjacent sections with false colour coding confirms the oligodendrocytic identity of Plexin-A1-positive cells in the white matter. Arrowheads indicate examples of double-stained oligodendrocytes. Scale bar = 10 μm.

signalling and cell migration, while it showed anti-tumour effect *in vivo* (Jacob *et al*, 2016). Strikingly, the addition of MTP-PlexA1 induced a similar disruption of NRP1/Plexin-A1 dimers, thereby confirming the inhibitory effect of the peptide (Fig 4D). Pre-incubation of the Oli-neu cells with MTP-PlexA1 ($10^{-7}$ M) cancelled Sema3A inhibitory effect by bringing migration up to the control condition without Sema3A (Fig 4E). This effect of MTP-PlexA1 was dose-dependent with a loss of efficacy from $10^{-9}$ M. Because of the toxicity of the vehicle (LDS) alone on Oli-neu cells (data not shown), we were not able to correctly evaluate higher concentrations of the peptide. However, because a maximal effect was obtained with $10^{-7}$ M we chose this concentration to define the dose for *in vivo* evaluation consistently with previous studies (Jacob *et al*, 2016).

**MTP-PlexA1 favours oligodendrocyte differentiation in the presence of Sema3A**

Remyelination failure in MS lesions can result from the lack of OPC recruitment as well as inhibition of OPC differentiation into mature myelinating oligodendrocytes. Our second *in vitro* functional test evaluated by RT–qPCR MTP-PlexA1 ability to increase a late oligodendroglial marker (*MBP*) expression during neural stem cell (NSC) differentiation. Plating of multipotent NSC onto PLO (poly-L-ornithine) with low growth factor concentration induces their differentiation into oligodendroglial, astrocytic glial and neural lineage. By adding triiodothyronine hormone and ascorbic acid into differentiation medium, we favoured oligodendroglial lineage. 100 ng/ml of Sema3A reduced by twofold the expression of *MBP* mRNA after

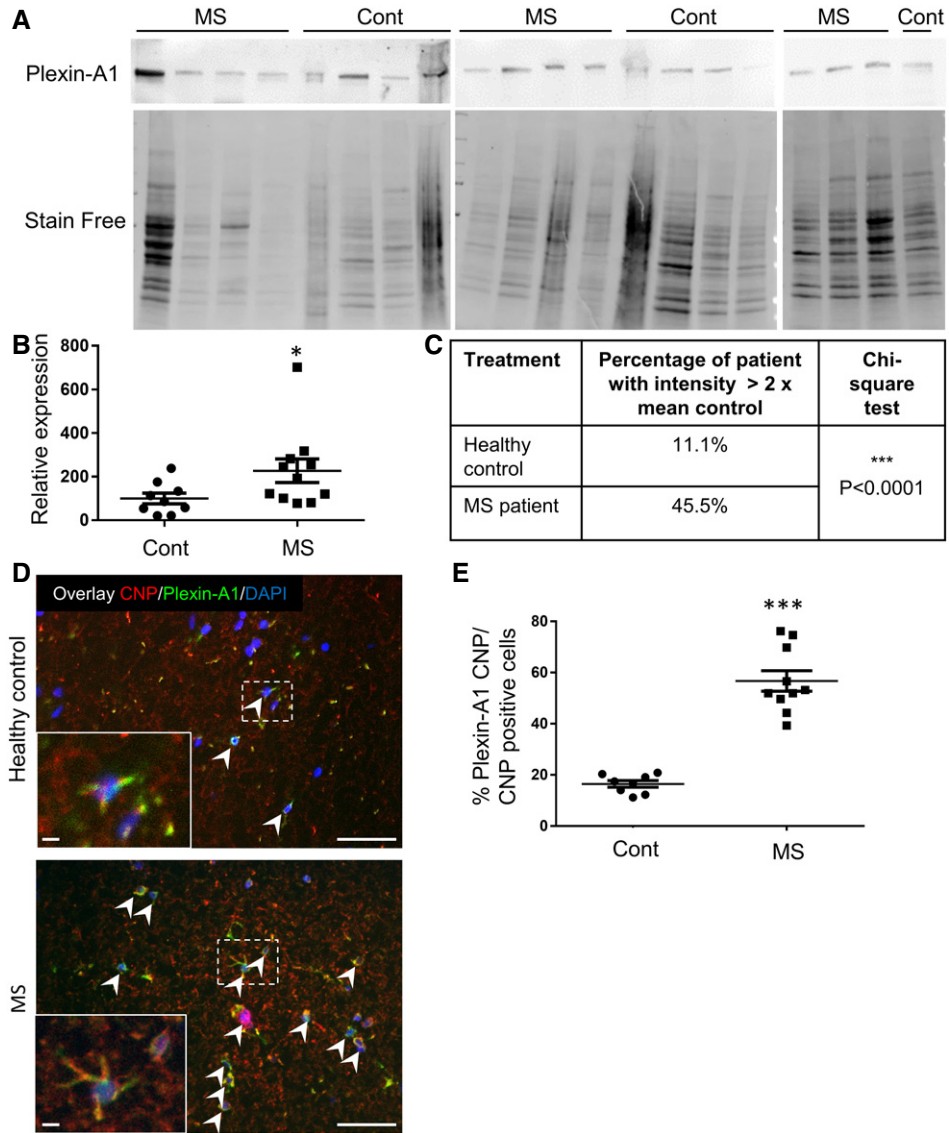

**Figure 2. Expression of Plexin-A1 in multiple sclerosis patients vs. healthy controls.**

A–C  Plexin-A1 immunoblotting analysis of brain samples of multiple sclerosis patients (n = 11) and healthy controls (n = 9). (A) Western blot revealed with anti-Plexin-A1 and stain-free method showing full protein content. (B) Relative expression normalized with full protein content (measured with stain-free method). Data are presented as mean ± SEM (Mann–Whitney, *P = 0.0167; n = 9 Ctrl and 11 MS patients). (C) Chi-square analysis of the proportion of patients with Plexin-A1 intensity > 2× mean control intensity.

D  Representative microphotographs illustrating the expression of Plexin-A1 in CNP-positive OL in healthy control or MS white matter samples (Plexin-A1: green, CNP: red). Arrowheads indicate oligodendrocytes (CNP) positive for Plexin-A1. Scale bar = 50 µm.

E  Quantification of the number of the CNP/Plexin-A1-positive cells in the white matter of control (HC) or MS autopsies. Data are presented as mean ± SEM (unpaired t-test, ***P = 0.0035; n = 9 Ctrl and 11 MS patients).

Source data are available online for this figure.

4 days of differentiation, whereas concomitant treatment with $10^{-7}$ M of MTP-PlexA1 brings back *MBP* mRNA expression to 1.2-fold of control condition without Sema3A (Fig 4F).

**MTP-PlexA1 exhibits no toxicity *in vivo***

Because of the expression of Plexin-A1 in adult neurons (Jacob *et al*, 2016), we had to check whether a chronic treatment with

MTP-PlexA1 could induce cognitive disabilities. We first evaluated the locomotion capability of vehicle (LDS) and MTP-PlexA1-treated animals that had received 1 µg/kg MTP-PlexA1 three times a week for 4 weeks in an open-field task (Fig 5A). No difference was found between the two groups, indicating that mice had the same exploration capacity. We then investigated mouse anxiety with an elevated plus maze (EPM) test. After quantification, groups exhibited no significant difference in open-arm exploration,

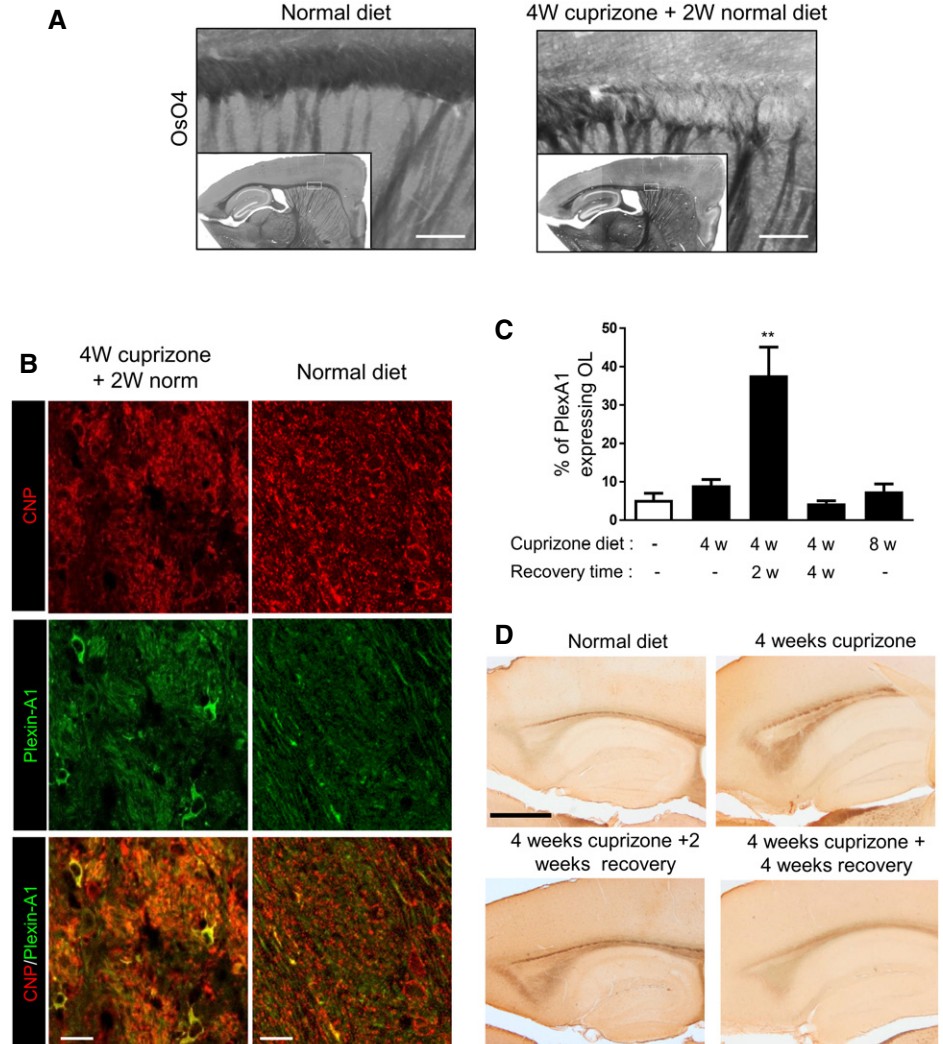

**Figure 3. Expression of Plexin-A1 and Sema3A in a model of adult demyelination–remyelination.**

Histological analysis of CNP/Plexin-A1-positive cells in animal receiving 4-week cuprizone diet (acute demyelination phase), 4-week cuprizone diet followed by 2-week normal diet (initiation of remyelination), 4-week cuprizone diet followed by 4-week normal diet (induction of total remyelination), and 8-week cuprizone diet (permanent demyelination).

A Representative examples of demyelination plaques seen by osmium tetroxide impregnation (OsO₄). Scale bar = 100 μm.
B Corresponding CNP/Plexin-A1 double staining showing Plexin-A1 expression in OL present at the lesion site. Scale bar = 10 μm.
C Quantification of the number of CNP/Plexin-A1-positive cells in the different experimental groups (w for weeks; data are presented as mean ± SEM, *n* = 3 mice per group in three independent experiments, 3–5 slices analysed per animal; ANOVA, **P = 0.0015).
D The expression of Sema3A is shown in adult brain at the level of hippocampus (sagittal sections) for control animals or animals receiving 4-week cuprizone diet (acute demyelination phase), 4-week cuprizone diet followed by 2-week normal diet (initiation of remyelination), and 4-week cuprizone diet followed by 4-week normal diet (induction of total remyelination). Scale bar = 1 mm.

demonstrating that MTP-PlexA1 has no measurable effect on plus maze-evaluated anxiety compared to vehicle (Fig 5B). Hence, we assessed the hippocampal function integrity with a spatial recognition task. Here again, the object discrimination capacity (determined through the recognition index) was identical in the two groups, thereby demonstrating no impact of MTP-PlexA1 on mouse cognitive functions (RI vehicle 0.36, RI MTP-PlexA1 0.35; Fig 5C and D).

To further address the innocuity of MTP-PlexA1, we also performed a blood sample analysis on four vehicle and four MTP-PlexA1-treated animals. White and red cell numeration showed no difference. Kidney and hepatic functions were also equivalent in the

two groups (Appendix Table S2). This lack of toxicity offered the possibility to test this administration schedule and dosing similar to what was previously performed to treat tumours (Jacob *et al*, 2016) in a model of demyelination to demonstrate the therapeutic potential of MTP-PlexA1.

**MTP-PlexA1 rescues corpus callosum myelination in demyelinating cuprizone murine model**

To investigate the therapeutic potential of MTP-PlexA1, we followed by MRI (DTI and T2WI) and histology the evolution of the white

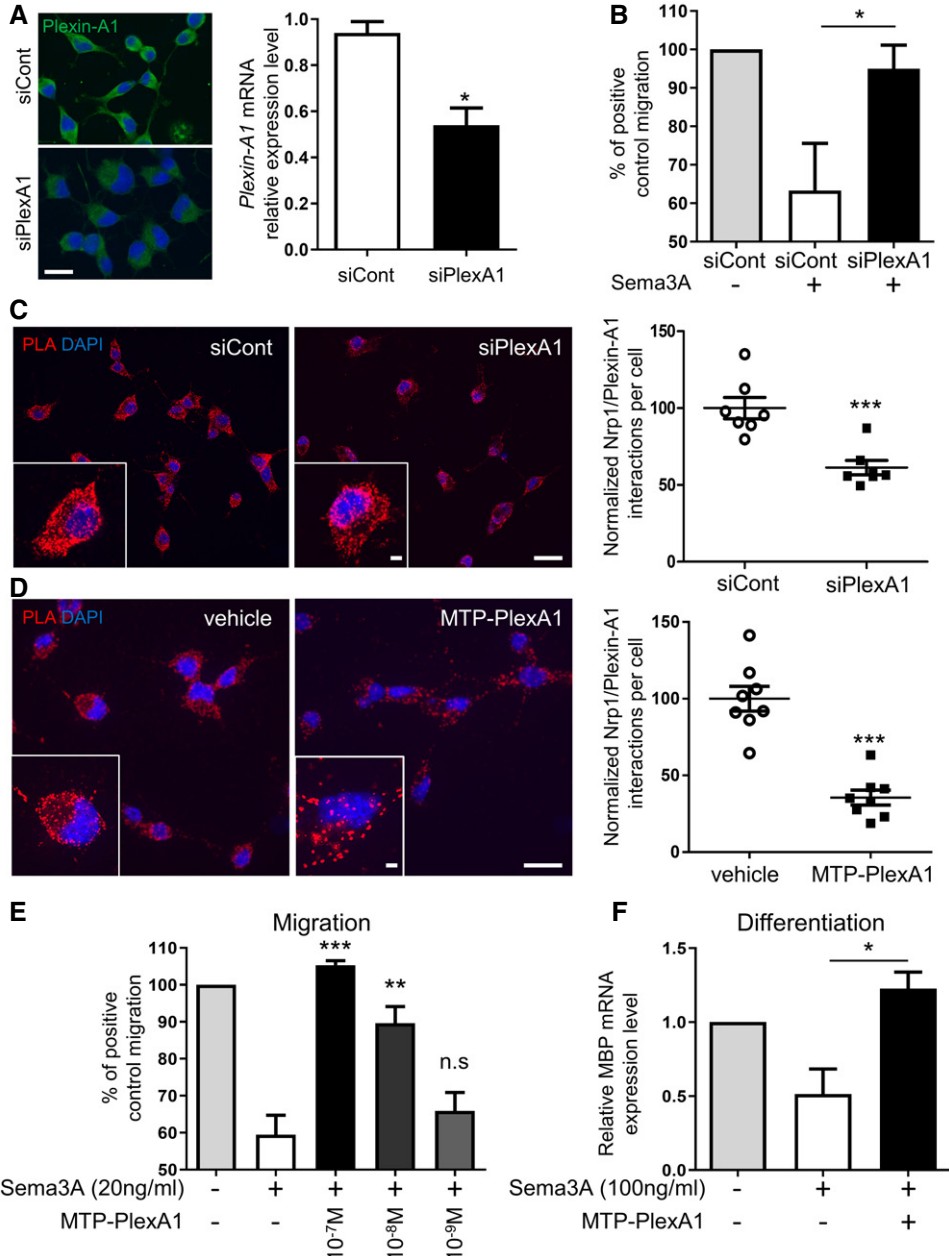

**Figure 4. Inhibition of PlexA1 rescues Sema3A negative effect on migration and differentiation.**

A   Cells were transfected with siRNA control or siRNA PlexA1. Downregulation of Plexin-A1 was validated by immunofluorescence staining with anti-Plexin-A1 antibody and by RT–qPCR analysis of Plexin-A1 mRNA expression normalized with GAPDH (data are presented as mean ± SEM, *n* = 3; Mann–Whitney test, *\*P* = 0.05). Scale bar = 10 μm.

B   Oli-neu cells were used for transfilter chemotaxis in response to 2% serum in the presence of Sema3A chemorepulsive (data are presented as mean ± SEM, *n* = 4 independent experiments, 1-way ANOVA and Kruskal–Wallis test, *\*P* = 0.0458).

C   Proximity ligation assay analysis was performed to quantify NRP1/Plexin-A1 dimers per cell treated with indicated siRNA. Representative microphotographs illustrating the different experimental conditions (data are presented as mean ± SEM, *n* = 7, Mann–Whitney test, *\*\*\*P* = 0.0006). Scale bar = 10 μm.

D   Proximity ligation assay analysis was performed to quantify NRP1/Plexin-A1 dimers per cell treated with vehicle or MTP-PlexA1. Representative microphotographs illustrating the different experimental conditions (data are presented as mean ± SEM, *n* = 8, Mann–Whitney test, *\*\*P* < 0.0001). Scale bar = 10 μm.

E   Oli-neu cells were used for transfilter chemotaxis in response to 2% serum in the presence of Sema3A chemorepulsive. Cells were pre-incubated with MTP-PlexA1 at indicated concentrations or vehicle alone. Results are expressed as a percentage of positive control migration, i.e. migration with 2% serum and without Sema3A and without chemoattractant (data are presented as mean ± SEM, *n* = 3 independent experiments, ANOVA and Bonferroni's multiple comparison test, *\*\*P* = 0.0025, *\*\*\*P* = 0.0005).

F   Expression of mature oligodendroglial marker MBP was analysed in murine neural stem cells (mNSCs) by RT–qPCR following 4 days of differentiation. Cells were concomitantly treated with Sema3A and MTP-PlexA1 or vehicle. Results are expressed relatively to differentiated cells without treatment (data are presented as mean ± SEM, *n* = 3 independent experiments, ANOVA and Kruskal–Wallis test, *\*P* = 0.0132).

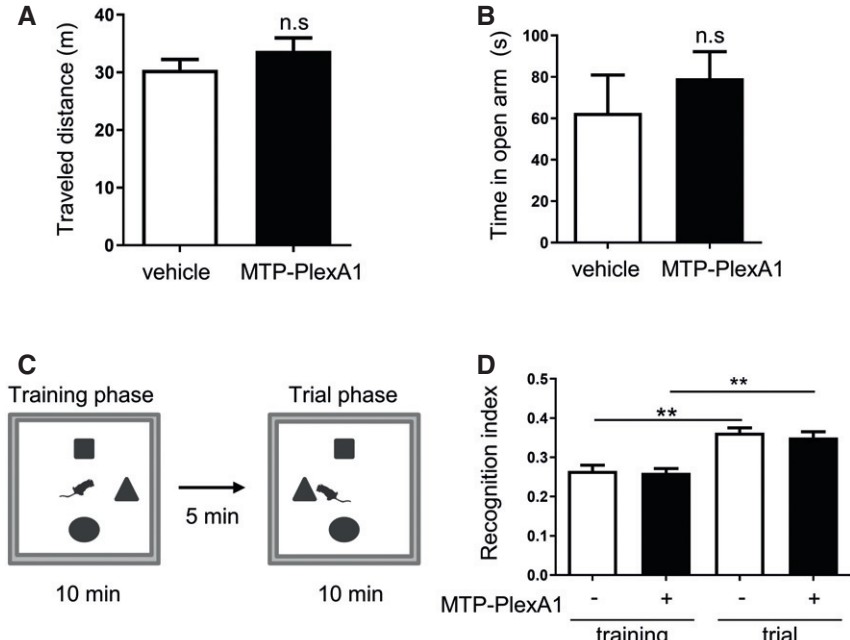

**Figure 5. Cognitive toxicity assessment.**

Mice were treated 4 weeks with vehicle or MTP-PlexA1 for behavioural tests.

A Determination of the global locomotion behaviour in the open-field task during 10 min (data are presented as mean ± SEM, n = 10 mice per group, Mann–Whitney test, n.s = not significant).

B Determination of the anxiety behaviour in the elevated plus maze with time spent in open arm during 10 min (data are presented as mean ± SEM, n = 10 mice per group, Mann–Whitney test, n.s = not significant).

C, D Spatial object recognition task measuring the novel object preference after training sessions. (C) Experimental procedure. (D) Recognition index = shifted object time/total exploration time (data are presented as mean ± SEM, n = 10 mice per group; Wilcoxon test, vehicle **P = 0.0020, MTP-PlexA1 **P = 0.0020).

matter using the cuprizone-induced demyelination–remyelination mouse model (Fig 6A). Averaged food intake monitoring confirmed equal consumption of the cuprizone diets in all experimental groups (Fig 6B). The analysis of T2WI signal intensity was used to evaluate the level of inflammation (signing the existence of a toxic effect of cuprizone), whereas DRAD (radial diffusion) signal increase was used as a readout of demyelination (Klawiter *et al*, 2011). We found that the T2WI signal intensity was identical in the two groups and the highest at W4 (123% of the baseline in control and 124% MTP-PlexA1). The T2WI signal progressively decreased in an equivalent manner for both groups at W6 and W8, thereby confirming an equal cuprizone-induced demyelination process (Fig 6C and D). To evaluate myelin damages, the t0 session was used as a normalization value for each mouse. DRAD was averaged to localize the demyelination areas in the corpus callosum (red to yellow pixels; Fig 6E). From t0 to W4, DRAD does not differ significantly from the baseline between vehicle (98%) and MTP-PlexA1 (97%). At W6, perturbation in the DRAD is observed for the vehicle group (110%) but not for MTP-PlexA1 group (103%), which almost stays at the baseline (Fig 6F). Strikingly, at W8, DRAD signals are significantly higher (two-way ANOVA, $P = 0.0196$) in the vehicle group (113%) than in the MTP-PlexA1 group (101%; Fig 6F). To confirm the apparent gain in myelination in MTP-PlexA1-treated mice, brains were collected at the end of the protocol to perform myelin histological staining. We performed Luxol fast blue staining on 6-μm-thick sagittal brain slices (Fig 7A). We

analysed Luxol staining in the splenium and adjacent half of the corpus callosum body, being the structures the most severely demyelinated by cuprizone. We expressed staining intensity as a percentage of healthy control. Mice fed 4 weeks with cuprizone and injected with vehicle during the whole protocol exhibited 67% of healthy control staining, whereas averaged staining of mice fed 4 weeks with cuprizone and treated with MTP-PlexA1 during the whole protocol reaches 100% of healthy control staining (Fig 7A). Histological pictures show a strong demyelination in the whole analysed area of demyelinated control. In mice treated with vehicle, intermediate demyelination appears and there was a clear lack of myelin compaction inside the splenium, which is not observed in healthy control and MTP-PlexA1-treated mice. This is further illustrated by immunostaining of PLP or staining with Fluoromyelin to visualize myelin content at the different stages (Fig 7A). Hence, we performed a proximity ligation assay on histological slices at the level of the lesions in the corpus callosum and splenium (Fig 7B). This analysis showed a dramatic loss of the number of NRP1/Plexin-A1 dimers (−50%, $P = 0.0027$) in treated animals compared to controls. While demonstrating that a sufficient amount of the peptide crossed the blood–brain barrier and reached the lesion sites, this analysis provides a demonstration of the mechanism of action of the peptide to exert its protective effects. Therefore, MTP-PlexA1 appears to rescue myelin in corpus callosum to a normal level despite cuprizone treatment by disrupting the signalling capability of Plexin-A1.

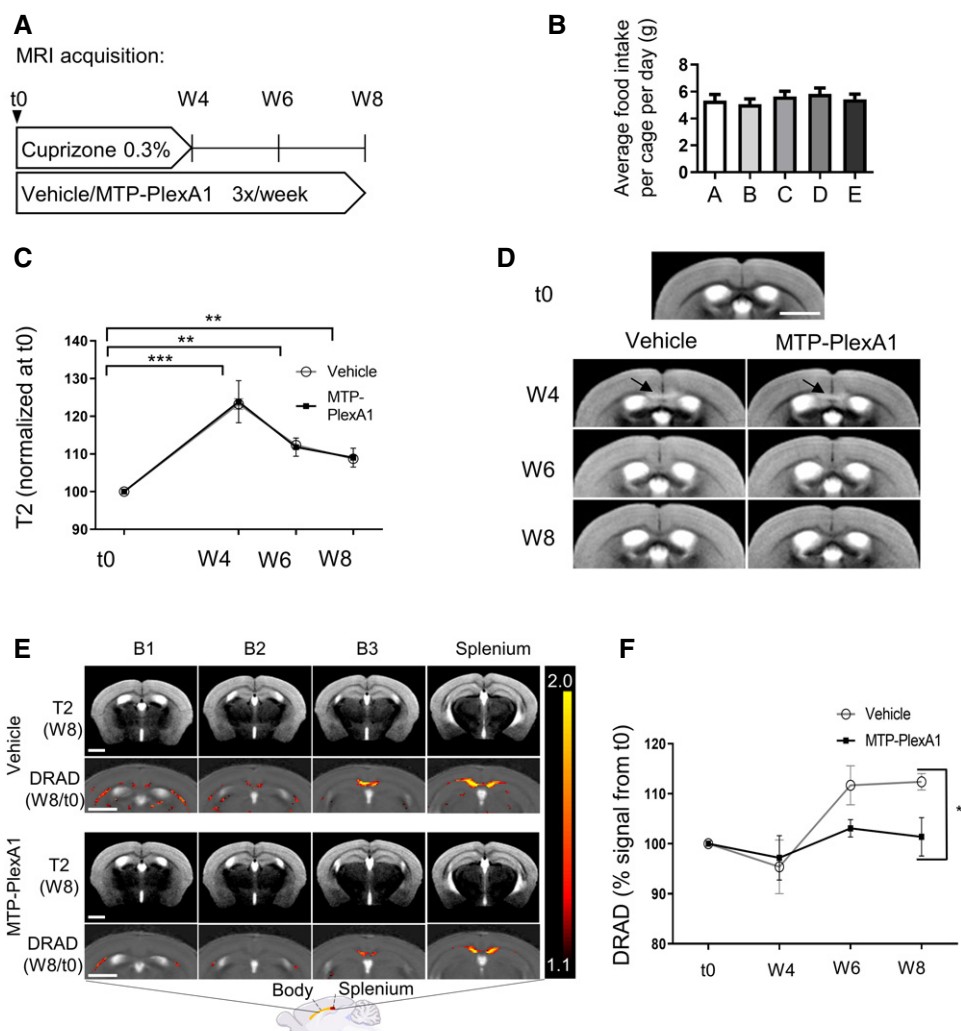

**Figure 6. Monitoring of cuprizone-induced demyelination.**

A   Schematic representation of the experimental protocol.
B   Averaged cuprizone-complemented food intake per cage (data are presented as mean ± SEM, n = 2 mice per cage).
C   Inflammation levels visualized by T2 signal intensity in the corpus callosum normalized by t0 values in the control group (n = 5) and treated group (n = 4). Data are presented as mean ± SEM. Two-way ANOVA multiple comparisons, ***P < 0.001, **P < 0.01 between groups.
D   Averaged T2 image obtained from signals in the vehicle and MTP-PlexA1 groups at W4, W6 and W8. Black arrows indicate inflammation at W4. Scale bar = 2.5 mm. Mice received 4-week cuprizone diet followed by 4-week normal diet with concomitant vehicle or MTP-PlexA1 injections (vehicle group n = 5 and MTP-PlexA1 group n = 4).
E   B1 (rostral side) to B3 (caudal side) correspond to part of the body of the corpus callosum. First line of each group is an average T2 image of the group at W8. Second line is the averaged DRAD maps obtained for each experimental group. Coloured pixels correspond to areas with a DRAD ratio W8/t0 between 1.1 and 2 corresponding to myelin loss. Scale bar = 2.5 mm.
F   DRAD MRI signal determined in the posterior half of the corpus callosum normalized by t0 values. Signal over 100% are correlated with myelin signal loss. Data are presented as mean ± SEM, vehicle n = 5, MTP-PlexA1 n = 4. Groups are significantly different at W8 (ANOVA, *P = 0.0196).

## Analysis of functional recovery with CatWalk Assay

MTP-PlexA1 allowing myelin recovery at the histological level, we looked for a positive effect *in vivo* on functional recovery. Locomotion disorders represent a functional disability found in MS patients and observed in demyelination models. We used CatWalk assay to measure cuprizone feeding effect on mice gait and to evaluate any curative potential of MTP-PlexA1. We established a 10-week experiment where mice were fed 6 weeks with

cuprizone, then treated 4 weeks with MTP-PlexA1 or vehicle for recovery. Cuprizone treatments induced alteration of temporal and kinetic parameters. Because the results obtained with forelimbs (Fig 8) and hindlimbs (see Appendix Fig S2) are identical at the same time points, they are described here without distinction. Stand Time, Swing and Step Cycle duration increased transiently after 2 weeks of curative treatment in mice receiving vehicle injections, whereas mice receiving MTP-PlexA1 presented no alteration of these parameters (Fig 8A). There was a

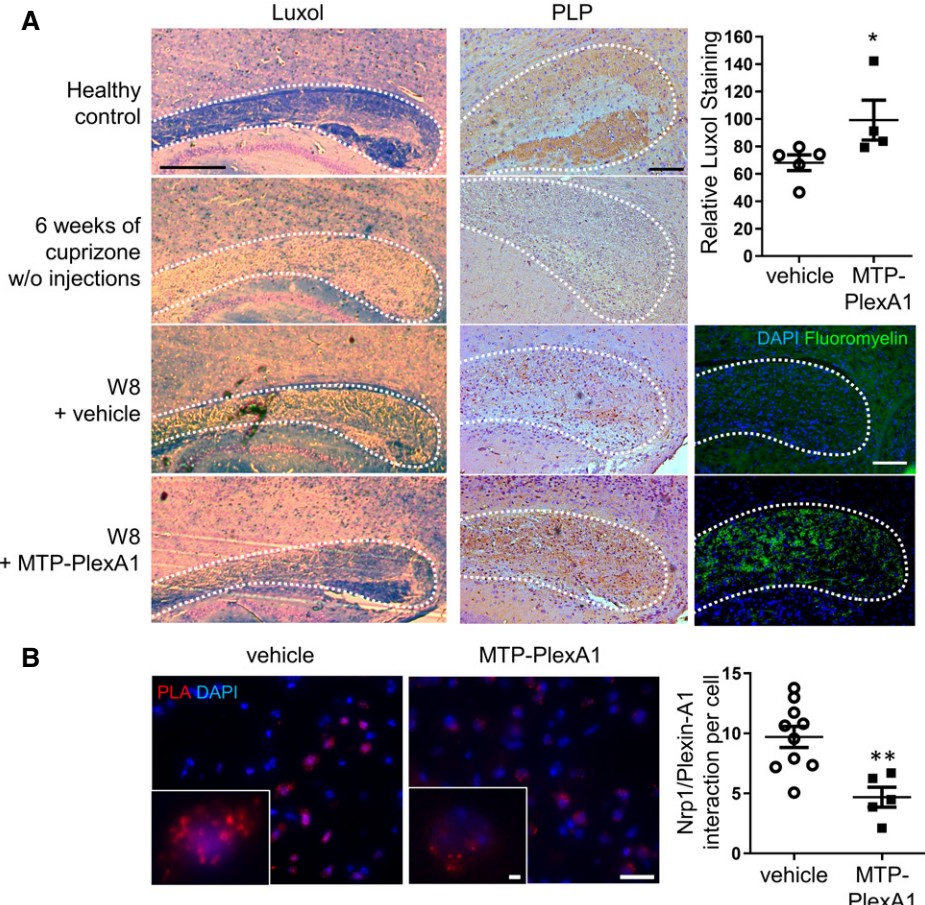

**Figure 7. Rescue of corpus callosum myelination after cuprizone-induced demyelination.**

Mice received 4-week cuprizone diet followed by 4-week normal diet with concomitant vehicle or MTP-PlexA1 injections.

A  Microphotographs showing the corpus callosum of mice fed with cuprizone and stained with Luxol fast blue, PLP and Fluoromyelin to visualize demyelination. Scale bar = 1 mm. White dot lines delineate the corpus callosum. Quantification of Luxol fast blue staining intensity is expressed as a percentage of healthy control (data are presented as mean ± SEM, vehicle *n* = 5, MTP-PlexA1 *n* = 4; Mann–Whitney test, *\*P* = 0.0159).

B  Proximity ligation assay analysis was performed to quantify NRP1/Plexin-A1 interactions per cells at the level of the corpus callosum. Representative microphotographs are presented. (data are presented as mean ± SEM, *n* = 9 microscopy fields of two representative vehicle-treated animals and five microscopy fields of two representative MTP-PlexA1-treated animals, Mann–Whitney test, *\*\*P* = 0.0027). Scale bar = 15 μm.

correlated significant Swing Speed decrease in mice receiving vehicle at 2 weeks of recovery (Fig 8B). No significant alteration occurred in Stride Length (Fig 8B). Therefore, 6 weeks of cuprizone feeding induced a transitory locomotor disorder, which appearance is prevented by MTP-PlexA1. Increased Stand Time indicated a longer postural phase, whereas increased Swing Duration without longer Stride Length indicated a slower and less effective propulsion phase. This altered propulsion could have different origins such as ataxia, muscular asthenia or spasticity, reflecting a large impact of cuprizone counteracted by a protective effect of MTP-PlexA1 treatment.

## MTP-PlexA1 exhibits protective effect in demyelinating EAE murine model

We extended our therapeutic approach to the EAE mouse model. This remitting–relapsing model is characterized by the appearance of an ascending paralysis. First, we validated expression of our therapeutic target at the peak of clinical score (reached 13 days after immunization). Indeed, Plexin-A1 mRNA is significantly more expressed in EAE mice compared to sham mice (Fig 9A). We next investigated the therapeutic potential of MTP-PlexA1 on the evolution of clinical scores and at the histological level with spinal cord myelin staining. Mice treated with vehicle only displayed first clinical signs 10 days after immunization. Then mean clinical score described a bell shape evolution with a maximum at day 13. Nonlinear regression showed that mean clinical score of mice treated from day 1 with 1 μg/kg MTP-PlexA1 was significantly lower compared to mice treated with vehicle (Fig 9B). This effect translated into a marked improvement of the myelin content at the lumbar spinal cord level in treated animals (Fig 9C). The Fluoromyelin analysis of the dorsal region showed +48% signal (*P* = 0.0124) and +41% in the lateroventral region (*P* = 0.0448; vehicle group *n* = 3; MTP-PlexA1 *n* = 3 for a total of 21 microscopy

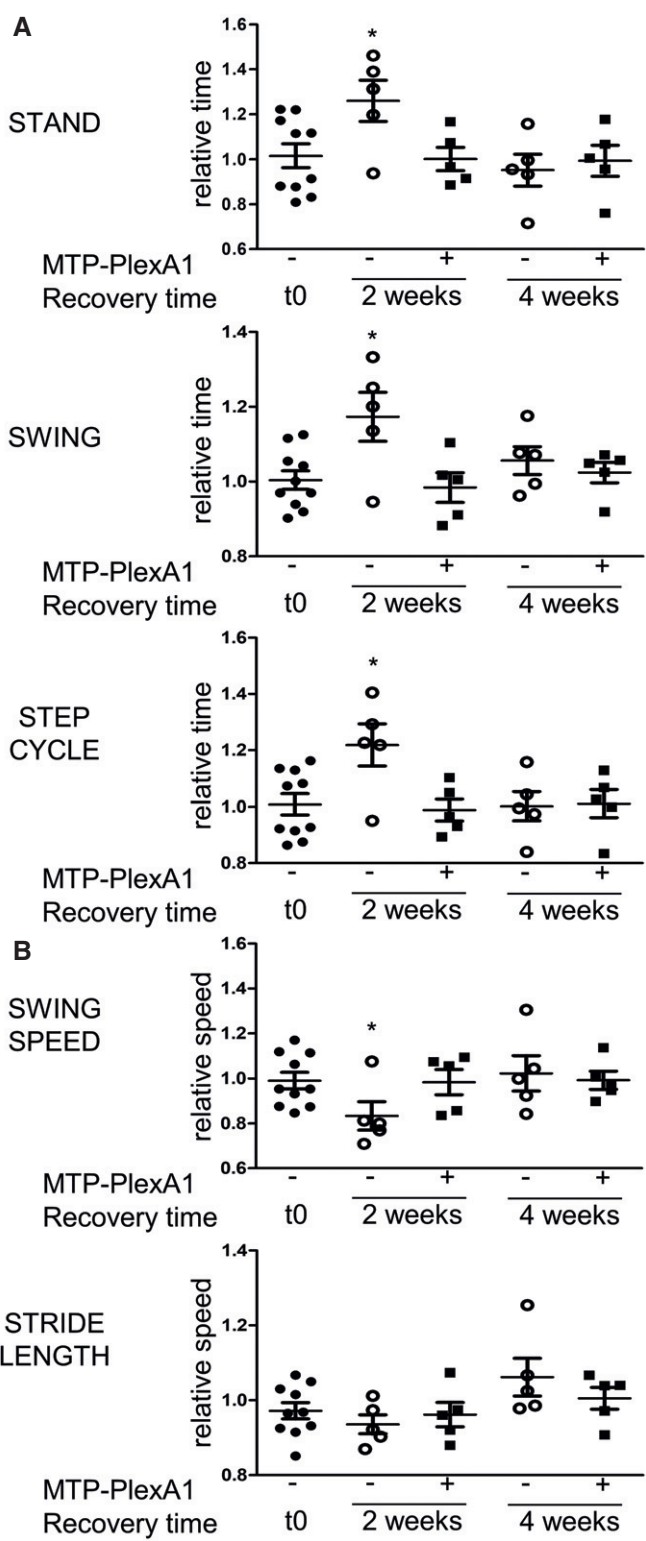

**Figure 8. Analysis of the functional recovery with CatWalk assay.**

Gait analysis of mice fed 6 weeks with cuprizone diet then receiving a curative treatment with MTP-PlexA1 or vehicle during additional 4 weeks of normal diet. All parameters are expressed relatively to the end of cuprizone treatment (representing the last time without deficits). Statistical significance is calculated towards values measured at the beginning of the experiment t0.

A   Temporal parameters (data are presented as mean ± SEM, $n = 5$ per group; Mann–Whitney, STAND, *$P = 0.028$, SWING, *$P = 0.028$, STEP CYCLE, *$P = 0.0127$).

B   Kinetic and spatial parameters (data are presented as mean ± SEM, $n = 5$ per group; Mann–Whitney, *$P = 0.028$).

day 10) and prolonged up to relapse phase. Strikingly, MTP-PlexA1-treated mice display no relapse in the next 28 days, whereas 28.6% of vehicle-treated mice display relapse with score ≥ 2 (see Appendix Fig S3). Importantly, the administration of MTP-PlexA1 had no impact on the inflammatory level as seen by the similar levels of circulating TNF-α measured by ELISA in all conditions at day 11 (Fig 9E).

## Discussion

Our expression data showed the expression of Plexin-A1 in mouse and human oligodendrocytes. This expression is higher in patients with MS compared to healthy controls. The *in vitro* studies demonstrated the capacity of MTP-PlexA1 to counteract Sema3A inhibitory effects on oligodendrocytes migration and differentiation. *In vivo* studies showed that administrations of MTP-PlexA1 every 3 days at the dose of 1 µg/kg induced beneficial therapeutic effects exemplified by myelin integrity signal recovery in DTI-MRI longitudinal follow-up of animals presenting cuprizone-induced demyelination. Histological examination of the brains at the end of the protocol showed complete restoration of myelin content in MTP-PlexA1-treated animals compared to only partial recovery in vehicle-treated animals. This myelin recovery is consistent with the functional recovery observed for MTP-PlexA1-treated animals for which five independent parameters of gait efficiency were measured. Such a protective effect was also observed in the EAE model in which we found that MTP-PlexA1 had a positive impact on the myelin content of lumbar spinal cord. Hence, behavioural studies combined with blood analysis demonstrated the innocuity of the MTP-PlexA1 peptides at the cognitive and biological levels.

Several studies indicate that Semaphorins play a role in OPC migration and OL process outgrowth (Piaton *et al*, 2009). Our results demonstrate that Plexin-A1, one of the major signalling receptors of Sema3A (Tamagnone *et al*, 1999; Tamagnone & Comoglio, 2000), is expressed in CNP-positive OL and overexpressed in case of induced demyelination. We also show that a Sema3A/Plexin-A1 signalling event triggers the inhibition of OL migration and differentiation.

Neuropilins and CRMPs are key components of class 3 Semaphorin signalling in OPC and OL (Ricard *et al*, 2000, 2001; Cohen, 2005). Other members of the Plexin family have also been suggested to act as regulators of Semaphorin signalling in OL. This is the case of Plexin B1 for the transduction of Sema4D (Moreau-Fauvarque *et al*, 2003) and that of Plexin A4 in OPC (Okada *et al*, 2007). Moreover, Sema4D-plexin-B1 interactions are critical for the pathogenesis

fields; Fig 9D). Moreover, the clinical scores of mice treated with 10 µg/kg MTP-PlexA1 were significantly improved compared to the 1 µg/kg condition. Hence, we challenged in a second experiment the therapeutic potential of MTP-PlexA1 when the treatment was administrated after first clinical symptoms appeared (in this case at

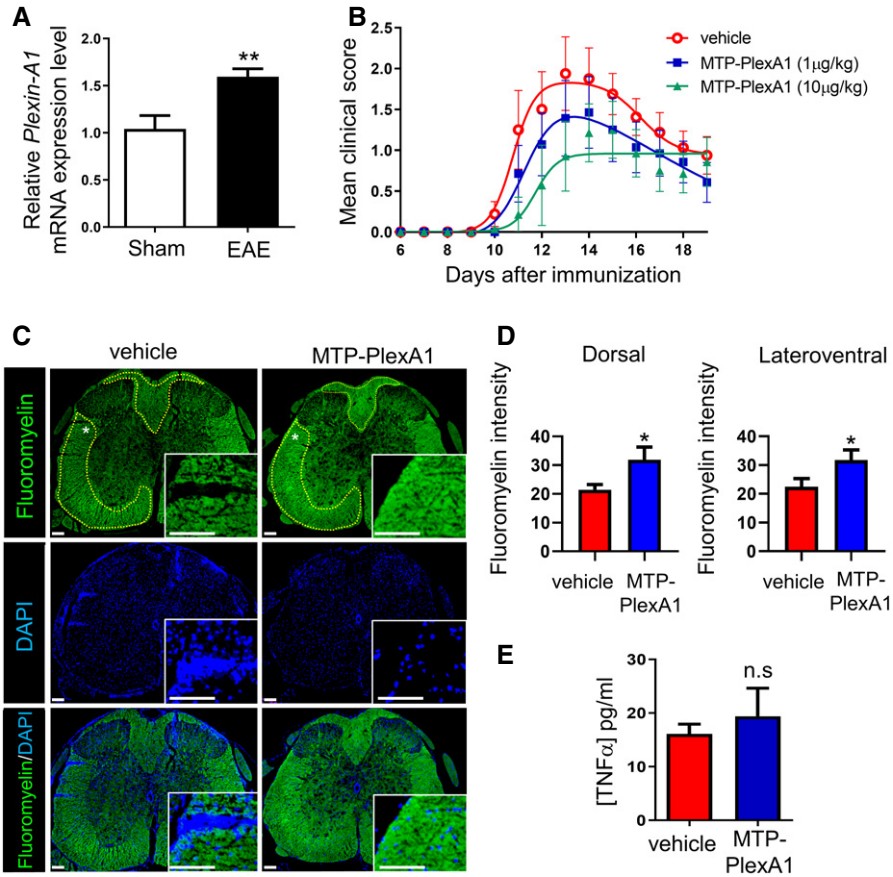

**Figure 9. MTP-PlexA1 reduces EAE severity.**

SJL female mice were immunized with PLP (+Pertussis toxin).

A   Expression of Plexin-A1 was analysed in the whole brains of PLP-sensitized mice exhibiting a clinical score superior to 1.5 (EAE) and sham mice by RT–qPCR. GAPDH was used as housekeeping gene to normalize gene expression (data are presented as mean ± SEM, n = 5 per group, Mann–Whitney test, **P = 0.0079).

B   Therapeutic treatment was administrated starting from day 1 (three times per week) consisting of vehicle alone, MTP-PlexA1 1 μg/kg or 10 μg/kg. Analysis of clinical signs of the disease. Data are presented as mean ± SEM, vehicle n = 6, MTP-PlexA1 (1μg/kg) n = 7, MTP-PlexA1 (10μg/kg) n = 7. Non-linear regressions (bell shape) are plotted and used for statistical significance (vehicle/MTP-PlexA1 (1 μg/kg), *P < 0.0001; vehicle/MTP-PlexA1 (10 μg/kg), *P < 0.0001; MTP-PlexA1 (1 μg/kg)/MTP-PlexA1 (10 μg/kg), *P = 0.0304).

C   Fluoromyelin and DAPI stain of 6-μm lumbar sections prepared from three mice treated with either the vehicle and three mice treated with MTP-PlexA1 (1 μg/kg) 11 days postinjection. White star shows the localization of the high magnification. Yellow dot lines delineate the area analysed for myelin content. Scale bar = 100 μm.

D   Fluoromyelin intensity analysis (data are presented as mean ± SEM, 3 slices per animal, n = 3 in both groups; dorsal Fluoromyelin content, *P = 0.0336; lateroventral Fluoromyelin content, *P = 0.0479; Mann–Whitney test).

E   Inflammation status of mice determined by ELISA for TNF-α 11 days postimmunization (data are presented as mean ± SEM, n = 4 in both groups; ns, Mann–Whitney test).

of EAE (Okuno *et al*, 2010). Our data now show that Plexin-A1 expression spatially and temporally fits with a role for Sema3A signal transduction in OL. Plexin-A1 expression significantly increased when provoking cuprizone-induced demyelination–remyelination process. In this situation, Plexin-A1 expression was only detected during the phase of acute demyelination during which we also found transient expression of Sema3A in the lesion sites mirroring what was previously shown in MS human samples (Williams *et al*, 2007), predominantly in the posterior third of the corpus callosum. This strongly supported a role of Plexin-A1 in the transduction of the Sema3A-induced migratory inhibition preventing remyelination. Indeed, we found *in vitro* that the silencing of Plexin-A1 using RNAi or the blockade with an antagonist peptide abolished the Sema3A-induced inhibition of OL migration. Interestingly, the inhibitory effect of Sema3A on OPC differentiation was

also counteracted when blocking Plexin-A1. This suggests that the beneficial effect seen *in vivo* with MTP-PlexA1 is related both on a pro-migratory effect and on a differentiation effect.

To block Plexin-A1, we have used a synthetic peptide mimicking its transmembrane domain. This type of antagonist previously described for other targets such as NRP1 (Nasarre *et al*, 2010; Arpel *et al*, 2016), HER2 (Arpel *et al*, 2016) or Eph-A2 (Sharonov *et al*, 2014) has been shown to act by the direct modulation of Plexin-A1 homodimerization or heterodimerization with Neuropilin-1. Long-lasting effects were described on brain tumour growth when the peptide was administrated every 3 days *in vivo* (Jacob *et al*, 2016). We now observed therapeutic effects of MTP-PlexA1 in the demyelination models when administrated with the same schedule and same dosage. While the exact fraction of peptide reaching the lesions remains to be determined, the efficient protective effects

observed in two different models of induced demyelination suggest that the required therapeutic dose is in the range of µg/kg. A proximity ligation assay indeed demonstrated a significant decrease in NRP1/Plexin-A1 dimers in brain slices prepared from treated animals. This assay confirmed a direct impact of MTP-PlexA1 on the signalling of Plexin-A1 in the lesion sites. However, membrane targeting peptides exhibit no specificity towards OL cells and Plexin-A1 is also expressed in adult neurons. While previous studies in cancer settings showed no obvious deleterious effects after 1-month treatments, we performed here series of behavioural experiments to challenge whether the peptide would or would not impair cognitive functions in a context of chronic administration. We found no effect on cognitive performance after 4-week treatments in three different tests. This was also correlated with no sign of hematologic or metabolic alteration. Thus, this antagonist peptide could be seen as an interesting drug candidate in the MS context. A translation to humans should be facilitated by the 100% homology between the mouse and human Plexin-A1 transmembrane domain sequences. Moreover, our comparative analysis of white matter from nine healthy controls or 11 MS patients revealed a marked increase in Plexin-A1 expression in MS. Future investigations will explore the dynamic of Plexin-A1 expression along the disease evolution to better identify the right therapeutic window while treating patients with anti-inflammatory medications. Hence, because current treatments of MS are essentially combating inflammation but are not repairing lesions, future studies will address whether a combination treatment associating anti-inflammatory molecules with Plexin-A1 antagonist creates all conditions for repairing MS lesions.

# Materials and Methods

## Cell culture

Oli-neu cells (Jung *et al*, 1995) kindly provided by Dr Trotter's laboratory were cultured at 37°C, with 5% $CO_2$, and expanded in Sato medium containing 1% horse serum (DMEM, with 0.2% (w/v) sodium bicarbonate, 0.01 mg/ml insulin, 0.01 mg/ml transferrin, 220 nM sodium selenite, 100 µM putrescine, 500 nM triiodothyronine, 520 nM thyroxine and 200 nM progesterone). Migration experiments were performed in Sato medium without progesterone, triiodothyronine and thyroxine on surfaces coated with poly-L-lysine.

E15 embryos were collected from pregnant C57BL/6 mice for brain dissection. Cells from cerebral hemispheres were mechanically dissociated and grown in suspension as neurospheres in Neurobasal-A medium supplemented with 2 mM L-glutamine, 2% B27, penicillin (100 U/ml), streptomycin (100 µg/ml), EGF (20 ng/ml) and bFGF (20 ng/ml). Differentiation was induced by plating neurospheres on poly-L-ornithine surface in Neurobasal-A medium supplemented with 2% B27, penicillin (100 U/ml), streptomycin (100 µg/ml), EGF (2 ng/ml), FGF (2 ng/ml), triiodothyronine (12.5 ng/ml) and ascorbic acid (50 µM).

## Peptide

We used as previously described (Jacob *et al*, 2016) the specific Plexin-A1 antagonist peptide corresponding to the transmembrane domain of Plexin-A1 (TLPAIVGIGGGGGLLLLVIVAVLIAYKRK, amino acid T1240 to K1268 according to UniProt entry P70206). The peptide (MTP-PlexA1) was synthetized and purified by Peptide Specialty (Heidelberg, Germany) by automatic peptide synthesis (Fmoc chemistry). Peptide purity estimated by RP-HPLC was more than 92% according to the manufacturer's indication. The peptide was solubilized in 72 mM LDS (lithium dodecyl sulphate) at a concentration of 1 mg/ml for stock solution and used at $10^{-7}$ M *in vitro* and administrated *in vivo* every 3 days at 1 µg/kg or 10 µg/kg.

## RNA interference

siRNA oligo control (AllStars Negative Control) and siRNA targeting mouse Plexin-A1 (Mm_Plxna1_5) were purchased from Qiagen and transfected into Oli-neu cells with INTERFERin (Polyplus-transfection).

## Cell migration assay

Migration of Oli-neu cells was performed in Transwell CIM-Plate 16 (8 µm pore size filter ACEA Biosciences, Inc.) with xCELLigence RTCA DP Instrument (ACEA Biosciences Inc.). Cells were pre-incubated 1 h with vehicle alone or MTP-PlexA1 ($10^{-7}$ M). The $1 \times 10^5$ cells were seeded in the upper chamber with 150 µl of medium. The bottom well contained 160 µl of medium supplemented with 2% foetal bovine serum for chemoattraction and 20 ng/ml Sema3A purified as previously described (Treinys *et al*, 2014) for repulsion. Analysis was performed after 6 h of migration according to the manufacturer's instructions. Data are expressed as a percentage of positive control migration, i.e. the migration of Oli-neu with 2% serum and without Sema3A.

## RNA analysis

Total EAE mouse brain RNA was extracted with TRIzol (Sigma) and analysed by reverse transcription–PCR. cDNA was amplified using the CFX Connect Real-Time PCR Detection System (Bio-Rad). A mix was done using the TaqMan™ Universal PCR Master Mix (Applied Biosystems) and the specific probe for Plexin-A1 (probe ID: Mm00501110_m1, Thermo Fisher). For normalization of gene expression, GAPDH (probe ID: Mm99999915_g1 Thermo Fisher) was used as housekeeping gene. MBP expression was analysed with SYBR Green Supermix (Bio-Rad) using following primers: GCCTGTCCCTCAGCAGAT and GCCTCCGTAGC-CAAATCC. The following programme was used: 95°C for 10 min, followed by 40 cycles of 95°C for 30 s, and polymerization at 60°C for 1 min. MBP expression was monitored after 4 days of differentiation in the presence of Sema3A with MTP-PlexA1 treatment or vehicle.

## Animal experimentation

All experiments were performed in accordance with the French animal protection laws and were approved by the Animal Care Committee of the University of Strasbourg (APAFIS number #8755-2017011817246097 and #9374-201605111128746v2). We used C57BL/6 female mice (Charles River Laboratories) for Cuprizone study. SJL/J female mice used for EAE were purchased from Janvier and Charles River. All mice were 8 weeks of age and fed

in a controlled environment (25°C) with free access to food and water and housed on a 12-h/12-h day/night cycle. Before any behavioural test, mice were put in experimentation room 1 h before every session to acclimatize to environmental conditions. After each trial, experimental set-ups were cleaned with 70% ethanol to eliminate odorant cues. All experiments were conducted in blind conditions with regard to mice treatments. At the end of protocols, mice were euthanized, and brains were collected for histological examination.

### Cuprizone-induced demyelination

Mice were housed in plastic cages pre-bedded corn (Innovive reference M-BTM-C8) with pre-filled acidified water bottle (Innovive reference M-WB-300A). Mice were paired-housed (one control with one treated), and cages were changed weekly. After 1 week of acclimation to environment, mice were fed *ad libitum* with a 0.3% cuprizone-containing diet (Safe Nutrition, E82220, Version 0373 A04) changed three times per week. Cuprizone diet consumption (three times a week) and body weight (once a week) were monitored during the whole protocol. Intraperitoneal administration of vehicle (LDS, 0.072 mM) or MTP-PlexA1 (1 μg/kg) was done three times per week.

Animals (five mice per condition: vehicle vs. MTP-PlexA1) subjected to DTI-MRI (Diffusion Tensor Imaging Magnetic Resonance Imaging), toxicity tests and histology analysis received 4 weeks of cuprizone diet, and those evaluated for locomotion recovery using the CatWalk system received 6 weeks of cuprizone diet followed by 4 weeks of normal diet *ad libitum*. For DTI-MRI experiments, animals received treatments (vehicle, MTP-PlexA1) from day 0 (start of cuprizone diet) and were imaged every 2 or 4 weeks (note that one mouse died during acquisition in the treated group at week 4). For CatWalk analysis (one session at t0, W4, W6, W8 and W10), animals received treatments (vehicle, MTP-PlexA1) for 4 weeks from the end of cuprizone diet (6 weeks).

### Preparation of brain sections

Animals were deeply anaesthetized before intracardiac perfusion of 4% formaldehyde at 4°C. Brains were then removed and postfixed in 4% (v/v) formaldehyde (FA) at 4°C for 2 h. After 24 h washing in TBS, frontal vibratome sections (70 μm thick) were performed and collected in TBS for immediate use or in Watson medium for −20°C conservation.

### Induction and assessment of active experimental autoimmune encephalomyelitis

SJL/J female mice (9 weeks old) were immunized with the kit developed by Hooke laboratories (EK-2120). Emulsion of PLP139-151 fragment (HSLGKWLGHPDKF) in CFA (complete Freund's adjuvant) was administered as four subcutaneous injections of 50 μl in the flank according to the manufacturer's protocol. Mice received 0.4 μg of pertussis toxin intraperitoneally on the day of immunization. Sham mice received only pertussis toxin and CFA without PLP. Clinical score was performed on 21 mice (vehicle $n = 8$; MTP-PlexA1 $n = 8$ for 1 μg/kg and $n = 5$ for 10 μg/kg). EAE was assessed clinically in blind conditions on a daily basis according to

the following criteria: 0, no disease; 1, decreased tail tone; 2, impaired righting reflex and partial hind limb paresis; 3, complete hind limb paralysis; 4, hind limb paralysis with partial forelimb paralysis; and 5, moribund or dead. To evaluate myelin integrity, we collected the spinal cord of three vehicle and three MTP-PlexA1 (1 μg/kg)-treated animals 11 days postimmunization. We also collected sera to determine TNF-α concentration with an ELISA kit (RD System MTA00B).

### Immunostaining

For immunohistochemistry on the US Biomax tissue array (BNC17011b SF40/41), we used adjacent tissue sections of 6 μm thickness. Sections were deparaffinized with toluene, then boiled with the antigen retrieval sodium citrate buffer (pH = 6) for 10 min. Sections were incubated overnight at 4°C with primary antibodies Plexin-A1 (ab23391, Abcam, 1:200), CNP (ab6319, Abcam, 1:500) or PLP (homemade, 1:500). Slides were then incubated with biotinylated secondary antibodies (Vector Laboratories, 1:200), amplified with the ABC Elite Vectastain kit and developed with the DAB kit from Vector Laboratories. Images of the two sections were false-colour-coded and overlaid in order to exemplify double-stained cells using ImageJ software.

Human cryosections of control and MS patients were prepared from frozen tissue blocks of white matter. All samples were obtained from the Netherlands Brain Bank, the Netherlands Institute for Neuroscience, Amsterdam (open access: www.brainbank.nl). All material has been collected from donors for or from whom a written informed consent for a brain autopsy and the use of the material and clinical information for research purposes had been obtained by the NBB in accordance with the principles set out in the WMA Declaration of Helsinki and the Department of Health and Human Services Belmont Report. Cryosection were co-stained with the same Plexin-A1 (1:200) and CNP (1:250) antibodies, before detection with Cy3 anti-rabbit and A488 anti-mouse secondary antibodies (Jackson ImmunoResearch, 1:300).

For Plexin-A1 expression study, brain sections were incubated overnight at 4°C under agitation with anti-Plexin-A1 antibodies and mouse anti-CNP antibodies before detection using Alexa488-conjugated anti-rabbit IgG and Cy3-conjugated anti-mouse IgG. The brain sections (prepared from three animals) were then mounted in Vectashield medium with DAPI. The percentage of Plexin-A1-positive oligodendrocytes was determined on at least five frontal sections at the level of the corpus callosum, the cortex and the striatum. Figure 3A shows Plexin-A1 expression in the corpus callosum, while quantification reflects averaged expression in the three brain regions.

### Western blot

Proteins were extracted in a PBS/Triton 1% buffer with protease inhibitors (Sigma) using the Minilys system (Bertin). Proteins were resolved in a 4–20% SDS–PAGE gel and transferred onto a nitrocellulose membrane (Trans-Blot Turbo System, Bio-Rad). The blots were soaked in blocking solution (TBS/0.1% Tween/5% milk) for 1 h at RT. Primary antibody (Plexin-A1 PA5-77697; Invitrogen, 1:1,000) was incubated overnight at 4°C. After several washes (three times for 5 min, TBS/Tween 0.1%), secondary antibody (anti-rabbit

HRP; Bio-Rad, 1:3,000) was incubated for 1 h at RT in TBS/1% Tween/5% BSA. The revelation step was performed using Clarity ECL Blotting Substrates (Bio-Rad) according to the manufacturer's instructions. Images of the immunoblots were acquired and analysed thanks to Chemidoc Touch Imaging System (Bio-Rad) and normalized with the stain-free method.

### Proximity ligation assay

Cells were seeded on Lab-Tek Permanox slides overnight, and then treated with appropriate peptide $10^{-7}$ M for 1 h or transfected with the appropriate siRNA for 48 h. After fixation with 1% PFA for 10 min, slices were permeabilized with PBS/0.1% Triton X-100. Primary antibodies (NRP1 from Evitria, 1:500; and Plexin-A1 from Abcam, ab23391, 1:200) were incubated overnight at 4°C in PBS. The proximity ligation assay was then performed according to the manufacturer's recommendations with the "detection orange" kit (Sigma). Quantification of the dots was performed using ImageJ software.

### Myelin staining

Plaques of demyelination (Matsushima & Morell, 2001) were observed at the histological level by the mean of osmium tetroxide ($OsO_4$) impregnation.

For histological analysis, brains were fixed with 10% FA, dehydrated and embedded in paraffin. Sagittal slices of 6 μm were cut with microtome. Luxol fast blue staining (Abcam) was performed according to the manufacturer's instructions. Images were collected with Leica microscope (LEITZ DMRB; equipped with AxioCam Zeiss) using ×1.6 objective. Images were processed with ImageJ software (National Institutes of Health) as follows. Red channel was extracted from RGB pictures and grey values inverted to get myelin signal. For each slice, the relative Luxol fast blue staining intensity was calculated by dividing the mean intensity measured in splenium and adjacent half of corpus callosum body by the mean intensity measured in cortex. Mean intensity of each brain was calculated from at least three slices.

EAE spinal cord was stored in a 4% PFA solution overnight at 4°C and then placed in 15% sucrose–phosphate-buffered saline solution for 24 h before embedding in OCT (optimal cutting temperature compound). Fluoromyelin staining (Invitrogen ref F34651 diluted at 1:200) and DAPI staining were performed on 10 μm coronal cryosections of the lumbar region of the spinal cord. Slice images were collected on Hamamatsu Nanozoomer S60 (FITC 80 ms, DAPI 8 ms). Fluoromyelin intensity was measured using ImageJ software. Regions of interest were drawn manually, D region corresponds to the dorsal part of the spinal cord and VL to ventrolateral (vehicle $n = 3$; MTP-PlexA1 1 μg/kg $n = 3$; the analysis was performed on a total of 24 microscopic fields).

For PLP detection, 6-μm paraffin sections were dewaxed, unmasked with citrate buffer (Sigma) and incubated overnight with anti-PLP antibody (PLP Lp6, 1:500), followed by incubation with a biotinylated goat anti-rabbit antibody diluted at 1:200 for 1 h at room temperature. After washing, slides were incubated with ABC Vectastain amplification system (Vector Laboratories) and revealed with DAB (Vector Laboratories). Counterstaining was performed using haematoxylin (Sigma).

### MRI recording

MRI examinations were performed on 7/30 Biospec System (Bruker BioSpin, Ettlingen, Germany). Transmission was achieved with a quadrature volume resonator (inner diameter 86 mm), and a standard mice brain quadrature surface coil (~ 19 × 19 $mm^2$) was used for signal reception (Bruker BioSpin, Ettlingen, Germany). MRI experiments were executed with ParaVision 6.0.1 software. T2-weighted axial anatomical dataset was acquired using RARE sequence employed using 39 contiguous slices with 0.3 mm thickness and an in-plane resolution of 0.078 × 0.078 $mm^2$. Remaining parameters were as follows: matrix 256 × 256, TE = 26.5 ms, TR = 5 s, $N$ avg = 6, RARE factor = 8 and acquisition time: 16 min. DTI was performed using a diffusion tensor spin-echo echo-planar imaging (DTI-SE-EPI) sequence with the following parameters: four shots, $N$ avg = 2, TR = 3 s, TE = 31 ms, 30 directions, 6 $b$-values, $\Delta/\delta$ = 12/6 ms and 23 consecutive slices with resolution = 0.1 × 0.1 × 0.5 $mm^3$, acquisition time: 60 min. For MRI experiments, anaesthesia was induced with 2.0–2.5% isoflurane. Then, animals were fixed in an animal cradle (Minerve, France) using a tooth bar and ear bars for stable positioning of the head. During MRI experiments, the anaesthesia was maintained with isoflurane to stabilize the breath frequency around 80–90 bpm. Respiration was monitored using a pressure pad under the thorax. MRI acquisitions were done at t0, W4, W6 and W8.

### Image processing and analysis

The signal bias of T2WI induced by the surface coil was corrected with N4 bias correction (Advanced Normalization Tools, ANTs). Brains were automatically extracted and co-registered to a reference using FSL tools: BET and FLIRT (FMRIB Software Library, Oxford, UK). A set of regions of interest corresponding to corpus callosum and ventricles was designed manually and applied T2WI to extract the signal of the corpus callosum normalized by the signal of the ventricles using ImageJ (NHI, USA).

After interscan drift correction, we used FSL (FMRIB Software Library, Oxford, UK) to reconstruct DRAD maps as a readout of myelin integrity. Then, individuals' T2WI, volume without diffusion gradient, was co-registered to a reference using FLIRT (FMRIB Software Library, Oxford, UK) and the transformation matrix was applied to the corresponding maps. A set of regions of interest corresponding to corpus callosum was designed manually and applied on all maps to extract values using ImageJ (NHI, USA).

To illustrate the myelin loss, we calculated the mean ratio of DRAD signal at W8/DRAD signal at t0 for each group. Pixels exhibiting ratio values > 1 (corresponding to demyelination) were selected using a mask before false colour coding and overlaid on the raw DRAD images (ImageJ FIJI).

### CatWalk assay

Gait analysis was performed using the CatWalk XT Automated Gait Analysis System (Noldus Technology Company) according to the manufacturer's instructions. Here are definitions of the parameter studied:

(i) Stand (s) or Stance Phase is the duration in seconds of contact of a paw with the glass plate. (ii) Swing (s) or Swing Phase is the duration in seconds of no contact of a paw with the glass plate. (iii)

Swing Speed is the speed (distance unit/second) of the paw during Swing. Swing Speed = Stride Length/Swing. (iv) Stride Length is the distance (in distance units) between successive placements of the same paw. Calculation of Stride Length is based on the *x*-coordinates of the centre of the paw print of two consecutive placements of the same paw during Max contact and taking into account Pythagoras' theorem. (v) Step Cycle is the time in seconds between two consecutive Initial Contacts of the same paw.

Values are expressed relatively to week 6 of the protocol corresponding to the last time without measurable deficit. Statistical significance is calculated by comparison with starting time of the experiment (t0).

### Elevated plus maze

The EPM has been used to assess anxiety in animals as proposed for pharmacological agents (Walf & Frye, 2007). In brief, we used a plus-cross-shaped apparatus, with two open arms and two arms closed by walls linked by a central platform 50 cm above the floor. Mice were individually put in the centre of the maze facing an open arm to explore the maze for a duration of 10 min. The time spent in the open arm, number of rearing, heading and stretch attend posture were used as an anxiety index. All parameters were measured using a Video Tracker software (Anymaze).

### Open field

The locomotion of animals treated for 4 weeks with MTP-PlexA1 or vehicle was determined in an open-field test where mice are placed individually in a polyvinyl plastic square open field. Mice that were freely moving and the travelled distance were measured using a Video Tracker software (Anymaze).

### Spatial recognition task

Mice were subjected to 10-min exploratory sessions in an open field enriched with three different objects. Five minutes later, one object was placed to the opposite side and the same mouse was introduced in the open field for a second exploratory session. The discrimination index was determined as the ratio of the time spent to explore the displaced object over the time spent to explore not displaced objects.

### Blood analysis

Blood analysis of eight mice (four vehicles and four treated) treated three times per week during 4 weeks with MTP-PlexA1 (1 μg/kg) or vehicle (0.072 mM) was conducted at the Clinical Chemistry and Hematology Platform of Institut Clinique de la Souris (ICS, Strasbourg, France) according to their laboratory routines.

### Statistical analysis

Graphs were produced in Prism 5 or 7 (GraphPad), and data are presented as mean ± SEM. Comparisons of one factor between two groups were analysed by unpaired *t*-test for normal distribution or Mann–Whitney test otherwise. Multiple comparisons were analysed with ANOVA and appropriate postanalysis. Significance was

**The paper explained**

**Problem**

There is currently no treatment to repair the myelin sheaths lost in multiple sclerosis. This crucial step is, however, mandatory to expect full recovery of patients benefitting from anti-inflammatory treatments used to jugulate the progression of the disease. One strategy to achieve remyelination is to counteract the inhibitory molecules accumulated in the lesion sites, thereby preventing spontaneous remyelination by blocking the arrival of oligodendrocytes for repair.

**Results**

In this study, we show that counteracting the Sema3A inhibitory molecular barrier by targeting its receptor Plexin-A1 favours remyelination. The use of a specific peptidic Plexin-A1 antagonist cancelled the anti-migratory and anti-differentiation effects of Sema3A by disrupting the heterodimer NRP1/Plexin-A1 required to trigger signalling. *In vivo*, the chronic administration of the peptide was remarkably tolerated and showed marked improvement in the myelin content and locomotor functions in two different animal models of MS.

**Impact**

These results demonstrate that the disruption of the Sema3A repulsive functions allows normal myelinating cells to exert their spontaneous remyelinating capacity. This opens unprecedented therapeutic opportunity by combining this novel approach with anti-inflammatory drugs to envision a possible curative therapeutic option.

accepted for values where $*P < 0.05$, $**P < 0.01$ and $***P < 0.001$. The number of animals per group for *in vivo* experiments was determined using the LaMorte (Boston University Medical Center) sample size calculation method with an anticipated effect of $-30\%$ with $\alpha = 0.05$ and a power of 0.95. Grubb test analysis was performed to exclude significant outliers in an experimental group. Animals were randomly assigned to experimental groups and assessed in blind conditions with regard to treatments. For animal studies, only one animal dying during the procedure (e.g. anaesthesia accident in the group MTP-PlexA1 in DTI-MRI experiment described in Fig 3) was excluded from the analysis. With the exception of Fig 2, variance is similar between groups statistically compared. In this particular Fig 2, we applied Welch's correction.

**Expanded View** for this article is available online.

### Acknowledgements

This work was supported by INSERM, ACI JC (#5327), Labex Medalis (MEDALIS-DBA-2016) and SATT Conectus (PLEXREMYEL) to Dominique Bagnard. The authors wish to thank Aurore Loeuillet for help with ELISA and Erwan Grandgirard for help with microphotograph acquisition at the IGBMC platform.

### Author contributions

FB performed functional *in vitro* assays. FB & LDP-V performed *in vivo* experiments and analysed data. GR, CS, LAM & LJ performed immunocytochemical experiments and analysed data. CS performed and analysed Western blot analysis. LM & AGM-N designed, supervised and analysed behavioural experiments conducted by LPV. CP designed and analysed MRI study performed together with LPV & FB. DBi performed qPCR experiments. VJ produced and scored in blind conditions EAE mice. MVH solubilized and validated peptide solutions. DBa designed and supervised the study,

analysed data and wrote the article. All authors provided critical reading of the article.

## Conflict of interest

The authors declare that they have no conflict of interest.

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
