## [Review Process File · EMBO Molecular Medicine]

Disruption of Sema3A/Plexin-A1 inhibitory signalling in oligodendrocytes as a therapeutic strategy to promote remyelination

Fabien Binamé, Lucas D. Pham-Van, Caroline Spenlé, Valérie Jolivel, Dafni Birmpili, Lionel A. Meyer, Laurent Jacob, Laurence Meyer, Ayikoé G. Mensah-Nyagan, Chrystelle Po, Michaël Van der Heyden, Guy Roussel and Dominique Bagnard.

Review timeline:

Submission date:	24 th January 2019
Editorial Decision:	15 th March 2019
Revision received:	22 nd June 2019
Editorial Decision:	5 th August 2019
Revision received:	23 rd August 2019
Accept:	4 th September 2019

Editor: Celine Carret

Transaction Report:

1st Editorial Decision

15th March 2019

Thank you for the submission of your manuscript to EMBO Molecular Medicine. We have now heard back from the three referees whom we asked to evaluate your manuscript.

As you will see from the reports below, the referees find the topic of your study of potential interest. While both reports highlight the interest of the paper, many issues are found, such as: missing controls and appropriate details/explanations, antibodies and results should be validated, n of mice should be indicated and increased when suggested, statistics should be defined, many imaging issues should be addressed. Referee 2 also mentions that an EAE mouse model should be used to better reflect the clinical relevance of MS, which is an important aspect of the work for us.

While it is clear that publication of the paper cannot be considered at this stage, given these overall evaluations I would be open to the submission of a revised manuscript. I must stress however, that the referee concerns must be fully addressed and that acceptance of the manuscript would entail a second round of review. I would add that it is particularly important that all of their suggestions [note that this would include the additional validation work on an EAE mouse model] are taken on board, as we cannot consider its publication otherwise.

REFEREE REPORTS

Referee #1 (Comments on Novelty/Model System for Author):

The models systems need to be more rigorously used.

Referee #1 (Remarks for Author):

Biname and colleagues report on experiments evaluation Plexin-A1/Sema3A signals in white matter and the targeting of this signaling to promote remyelination. They study human autopsy multiple sclerosis and control brain cases and mice as well as a mouse oligodendrocyte cell line. The study Plexin-A1 and Sema3A in human brain tissue and in developing mouse brain by immunohistochemistry and immunofluorescence. They use an adult mouse model of toxin-induced demyelination/recovery (remyelination) to localize Sema3A and Plexin-A1. This mouse model was also used to test Plexin-A1 antagonistic peptide as a therapeutic with clinical neurologic and MRI outcomes, after cell culture experiments showed the antagonistic activity of this peptide. The authors conclude that disruption of brain Sema3A/Plexin-A1 signaling could be therapeutic in white matter disease.

This study has several strengths. The topic is important and clinically significant. The overall experimental design using human MS autopsy tissue, cell culture, and a mouse model of demyelination was a reasonable and logical structure. The MRI assessment was good and refreshing because anatomically-matched levels were shown in the figures and the number of mice used were precisely identified. All experiments should use the general MRI design and thoroughness for data presentation.

This study has important weaknesses in the experimental design and explanation of experiments.

1. The human immunohistochemistry (Fig. 1) and immunofluorescence (Fig. 2) need negative controls for the Plexin-A1 and Sema3A staining.
2. The validation of the Plexin-A1 and Sema3A antibodies for use in human tissue needs to be shown.
3. The apparent changes in Plexin-A1 and Sema3A in MS cases need to corroboration by western blotting.
4. In Figure 2A, only 2 controls cases were used. This N needs to be increased.
5. In Figure 2B, higher magnification insets need to be shown for the labeled cells.
6. Figure 3. 3A needs a low magnification view showing neuroanatomical perspective. A DAPI stain for the nucleus would be useful.
7. Figure 3B. The numbers of mice used for each group needs to be identified. Statistical analysis needs to be shown.
8. Figure 4. A. The anatomical levels compared seem to be different. Lower magnification views need to be shown precisely identifying where the pictures were taken.
9. Figure 4. B. The CNP seems to have much background nonspecific staining. Negative controls need to be shown.
10. Figure 4 C. The number of animals used in each group needs to be identified in the figure legend.
11. Figure 5. This figure needs vast improvement. It would be good to compare the same anatomical levels in all 4 panels. All images are very different and the medial-lateral plane of section.
12. Figure 8D. The T2 signal intensity is too bright to make out any treatment group-related changes.
13. Figure 9A. Matching T2 images without pseudo color would be useful to show.
14. Figure 9C. The histology is very poor.

Referee #2 (Comments on Novelty/Model System for Author):

While the cuprizone model is appropriate to study the role of the Sem3A pathway during deyelination/remyelination, it does not reflect other important aspects of MS pathology and pathogenesis. The potential of interfering with this inhibitory pathway should be assessed in the EAE model of MS.

Referee #2 (Remarks for Author):

In this manuscript, Biname and colleagues sought to investigate the potential of a new MS drug. They performed proof-of-mechanism work in human MS tissue and the cuprizone mouse model, showing that Sem3A/PlexinA1 signalling negatively contributes to remyelination after determining that these antigens are expressed in the oligodendrocyte lineage. The preclinical benefit of an PlexinA1 antagonist was investigated in the cuprizone model. In an elegant in vitro model, the

authors could dissect that the antagonist action is beneficial for both cell differentiation and migration. This is an interesting study with promising preliminary data. Control experiments would substantially improve data validity and mechanistic insights. Moreover, syntax and some terminology need to be reviewed and some rephrasing is required.

My specific comments are outlined below and refer to page numbers of the pdf:

Major points:

1. While the cuprizone model is appropriate to study the role of the Sem3A pathway during demyelination/remyelination, it does not reflect other important aspects of MS pathology and pathogenesis. The potential of interfering with this inhibitory pathway should be assessed in the EAE model of MS.
2. Missing control experiments to evaluate the therapeutic potential of this pathway include RNAi (at least for in vitro models) to support the beneficial antagonist data; Additionally, the specificity for the PlexinA1 signalling-dependent mechanisms should be confirmed by experiments including a PlexinA1 agonist resulting in attenuated remyelination in the cuprizone model. Alternatively, virus-mediated over/underexpression of Sem3A/PlexinA1 could be performed.
3. Fig. 3C+D: Show representative data of MOG and NG2 stainings. MOG is a myelin marker and hardly stains the OL soma, making colocalization with other somatic markers challenging. Appropriate markers are e.g. ASPA or GSTPi.
4. Antagonist: How did the authors arrive at these doses? Please provide PK/PD data. Specifically dose-response data should be included in the models relevant for the current study. i.e. cuprizone model, OliNeu cells

Minor points:

1. Page 6, Reference "Mi": Delete "1"
2. Page 5, Results 1st paragraph: Rephrase: e.g. white matter oligodendrocytes
3. Scale bars in Fig. 1 are either confusing or completely missing
4. Page 6, CNP is a marker of mature OLs. Co-localization studies with OL lineage cell markers during development should use a pan-OL marker in addition to take this into consideration.
5. Page 7, "Moreover, similarly to what has been described in human samples of multiple sclerosis." - watch syntax; This is not a full sentence. Rephrase.
6. Fig. 6B, What mRNA levels are shown?
7. Page 10, "Results obtained with forelimbs and hindlimbs (see supplementary Fig. 2) were significantly not different at the same time points," - This doesn't make sense. Rephrase: e.g. did not differ
8. Generally: Use english decimal presentation "." instead of ","
9. Page 11, "maximal phase of myelination" - Rephrase: e.g. peak myelination
10. Fig. 1, Clearly state that the double immunohistochemical presentation maybe possible to inform co-localization because both antigens were detected in the same section. It should be noted that though that these data are not reliable as they were obtained employing non-confocal microscopy

Detailed response to reviewer's comments:

Referee #1 (Remarks for Author):

Biname and colleagues report on experiments evaluation Plexin-A1/Sema3A signals in white matter and the targeting of this signaling to promote remyelination. They study human autopsy multiple sclerosis and control brain cases and mice as well as a mouse oligodendrocyte cell line. They study Plexin-A1 and Sema3A in human brain tissue and in developing mouse brain by immunohistochemistry and immunofluorescence. They use an adult mouse model of toxin-induced demyelination/recovery (remyelination) to localize Sema3A and Plexin-A1. This mouse model was also used to test Plexin-A1 antagonistic peptide as a therapeutic with clinical neurologic and MRI outcomes, after cell culture experiments showed the antagonistic activity of this peptide. The authors conclude that disruption of brain Sema3A/Plexin-A1 signaling could be therapeutic in white matter disease.

This study has several strengths. The topic is important and clinically significant. The overall experimental design using human MS autopsy tissue, cell culture, and a mouse model of demyelination was a reasonable and logical structure. The MRI assessment was good and refreshing because anatomically-matched levels were shown in the figures and the number of mice used were precisely identified. All experiments should use the general MRI design and thoroughness for data presentation.

This study has important weaknesses in the experimental design and explanation of experiments.

1. The human immunohistochemistry (Fig. 1) and immunofluorescence (Fig. 2) need negative controls for the Plexin-A1 and Sema3A staining.

We appended to this detailed response additional figures for review only. The first figure is showing the different controls of antibodies used in this study. (Additional figure 1). This includes 1) omission of primary antibody to catch fluorescently-labelled secondary antibodies background 2) Tissue array with validated positive or negative tissues for Plexin-A1 and CNP and 3) western blot analysis with recombinant murine or human Sema3A proteins.

2. The validation of the Plexin-A1 and Sema3A antibodies for use in human tissue needs to be shown.

See above. (Additional figure 1).

3. The apparent changes in Plexin-A1 and Sema3A in MS cases need to corroboration by western blotting.

We were able to corroborate the overexpression of Plexin-A1 in western blot analysis of the white matter samples used for the immunocytochemistry study. Importantly, the collected autopsies showed important degradation as measured by Agilent Bioanalyser (see additional figure 2 for review only). However, a 150 kd band corresponding to a fragment of the Plexin-A1 extracellular domain could be detected in samples. The whole blots showing additional bands reflecting the degradation of samples are provided in additional figure 2. The selected 150kd band and corresponding stain free images (used for normalization and quantification of the signals) are shown in revised Figure 2. This result shows that on average, MS patients exhibit a two-fold increase of Plexin-A1 expression. Because the overexpression of Plexin-A1 was variable from one patient to another one, we also analyzed the proportion of patients exhibiting at least a two-fold increase of Plexin-A1 expression. This analysis confirmed that Plexin-A1 is significantly overexpressed in MS patients. The immunocytochemical analysis initially performed on the white matter provides results with a better resolution and confirmed at the cellular level that oligodendrocytes overexpress Plexin-A1 in MS patients.

We have not performed any western-blot analysis of Sema3A in MS patients. The data initially presented in Figure 2 (now supplementary Figure 1) were collected from a gene array profile published in Han et al., 2012 and reanalyzed by us with the GEO platform for Plexin-A1 and SEMA3A expression.

4. In Figure 2A, only 2 controls cases were used. This N needs to be increased.

Here again, this analysis was performed from the data set published by Han et al., 2012 and accessible from the GEO platform. Therefore, we cannot increase the number of cases. Considering the results obtained with our own samples at the protein levels (Figure 1 with immunohistochemistry and Figure 2 western blot and immunohistochemistry), we removed this gene array profile retrospective analysis from figure 2 and placed it as supplementary figure 1.

5. In Figure 2B, higher magnification insets need to be shown for the labeled cells.

Figure 2B (now Figure 2D) has been modified according to the suggestion. Insets showing high magnification of double positive (CNP/Plexin-A1) cells are now displayed.

6. Figure 3. 3A needs a low magnification view showing neuroanatomical perspective. A DAPI stain for the nucleus would be useful.

Figure 3 has been removed from the paper to focus the message on the Plexin-A1 expression in the disease context. The developmental expression pattern and electron microscopy analysis will be published elsewhere. For review only, we provide additional figure addressing anyway some of the remarks. Due to lack of time we were not able to replicate these experiments that were initially performed without DAPI staining and analyzed by confocal microscopy (some with Z-stack acquisitions). We provide herein selected microphotographs better illustrating the double staining in the different anatomical regions. Now additional figure 3 (for double staining with CNP and in the different brain regions), 4 (for double staining with O4) and 5 (for double staining with NG2 or MOG).

7. Figure 3B. The numbers of mice used for each group needs to be identified. Statistical analysis needs to be shown.

For the developmental expression, 3 mice of each age were collected. A minimum of 3 slices per anatomical regions were quantified.

8. Figure 4. A. The anatomical levels compared seem to be different. Lower magnification views need to be shown precisely identifying where the pictures were taken.

Low mag microphotographs are now provided in the revised Figure 4.

9. Figure 4. B. The CNP seems to have much background nonspecific staining. Negative controls need to be shown.

Negative controls were obtained by omitting primary antibodies (additional Figure 1). In all cases, high magnification observations were required to detect fluorescent signals. The images are Z-stack obtained with confocal microscopy. We provide additional illustrations in additional figure 3 including images with positive or negative Oligodendrocytes in the same fields in different brain regions. Of note, the tissues were disorganized and Oligodendrocytes misaligned in cuprizone treated animals. As seen in additional figure 1, the CNP antibody used is providing a specific staining of the white matter, hence, oligodendrocytes) as seen in the cerebellum with adjacent positive (white matter) and negative (grey matter) regions) while being completely negative in a malignant adrenal gland.

10. Figure 4 C. The number of animals used in each group needs to be identified in the figure legend.

The details on the number of mice and number of analyzed fields are provided in the method section. Indeed, in new figure 3 (ex-Figure 4C), 3 mice per condition (3 out of 5 per experimental group) from 3 independent experiments were collected for histological examination. A minimum of 3 slices (up to 5 slices) per animal were quantified. This corresponded to a total of 407 cells for control animals, 255 cells at 4W, 437 cells at 4W+2W, 383 cells at 4W+4W for and 309 cells W8. The statistical analysis shows a significant increase at 4W+2W compared to control ($p=0.0015$, ANOVA test followed by Holm-Sidak's multiple comparisons test).

11. Figure 5. This figure needs vast improvement. It would be good to compare the same anatomical levels in all 4 panels. All images are very different and the medial-lateral plane of section.

The cuprizone treatment is inducing lesions with variable intensity. We focus our analysis in the corpus callosum and the splenium being the regions displaying in our hands the most reliable lesions. However, the exact localization may vary from one animal to another one. To illustrate the expression of Sema3A in the different experimental conditions, we had selected the slices with the highest signal. This is the reason why the slices were not exactly at the same level. To further strengthen our results, we now repeated the experiments on additional slices and cases illustrating the dynamic expression of Sema3A and this, in a more similar anatomical region. Results are shown in revised figure 4.

12. Figure 8D. The T2 signal intensity is too bright to make out any treatment group-related changes.

The contrast and light intensity levels of the T2 images have been corrected for better display. The quantitative analysis which was performed from the unmodified original T2 images is unchanged. The statistical analysis we performed is now better illustrated in revised Figure 8. The two groups exhibit the same level of inflammation thereby suggesting same induction of demyelination. This inflammation signal is significantly increased for both groups at Week 4 and progressively diminishes at week 6 and 8.

13. Figure 9A. Matching T2 images without pseudo color would be useful to show.

Matching T2 images (being the average of all mice of each experimental group) for each group have now been added in Figure 9A.

14. Figure 9C. The histology is very poor.

To further illustrate the demyelination-remyelination process we conducted additional histological staining using Fluoromyelin and anti-PLP immunostaining. Figure 9C has been revised accordingly to include alternative illustrations of the myelin status in the different groups. This revised figure also display the results obtained with PLA (Proximity Ligation Assay) and demonstrating a significant disruption of the number of NRP1/PlexA1 dimers in treated animals.

Referee #2 (Comments on Novelty/Model System for Author):

While the cuprizone model is appropriate to study the role of the Sem3A pathway during deyelination/remyelination, it does not reflect other important aspects of MS pathology and pathogenesis. The potential of interfering with this inhibitory pathway should be assessed in the EAE model of MS.

Two additional experiments were conducted in the EAE model to confirm the therapeutic potential of blocking Plexin-A1

Referee #2 (Remarks for Author):

In this manuscript, Biname and colleagues sought to investigate the potential of a new MS drug. They performed proof-of-mechanism work in human MS tissue and the cuprizone mouse model, showing that Sem3A/PlexinA1 signalling negatively contributes to remyelination after determining that these antigens are expressed in the oligodendrocyte lineage. The preclinical benefit of an PlexinA1 antagonist was investigated in the cuprizone model. In an elegant in vitro model, the authors could dissect that the antagonist action is beneficial for both cell differentiation and migration. This is an interesting study with promising preliminary data. Control experiments would substantially improve data validity and mechanistic insights. Moreover, syntax and some terminology need to be reviewed and some rephrasing is required.

My specific comments are outlined below and refer to page numbers of the pdf:

Major points:

1. While the cuprizone model is appropriate to study the role of the Sem3A pathway during demyelination/remyelination, it does not reflect other important aspects of MS pathology and pathogenesis. The potential of interfering with this inhibitory pathway should be assessed in the EAE model of MS.

Two additional experiments were conducted in the EAE model to confirm the therapeutic potential of blocking Plexin-A1. In the first experiment the treatment was administered one day after immunization. In this case we found that MTP-PlexinA1 significantly reduces the severity of the disease in a dose dependent manner. This effect could be explained by a preservation of the myelin content in the spinal cord. Interestingly, while a QPCR-analysis showed an increase of Plexin-A1 expression in the brain of diseased animals, the monitoring of TNF α showed that the level of inflammation was similar in both groups thereby suggesting that MTP-PlexinA1 has no anti-inflammatory effect.

2. Missing control experiments to evaluate the therapeutic potential of this pathway include RNAi (at least for in vitro models) to support the beneficial antagonist data; Additionally, the specificity for the PlexinA1 signalling-dependent mechanisms should be confirmed by experiments including a PlexinA1 agonist resulting in attenuated remyelination in the cuprizone model. Alternatively, virus-mediated over/underexpression of Sem3A/PlexinA1 could be performed.

RNAi experiments were conducted in vitro on Oli-Neu cells. We obtained a significant 50% reduction of Plexin-A1 expression which was sufficient to restore an almost complete migration of cell exposed to a gradient of Sema3A. To address the question of the specificity we used a proximity ligation assay. This test allowed us to measure the relative number of NRP1/PlexA1 dimers in cells treated or not with MTP-PlexinA1. The results showed that indeed as previously described in cancer cells (Jacob et al., Oncotarget, 2016), MTP-PlexA1 is significantly disrupting the heterodimerization of NRP1 and Plexin-A1 in oligodendrocytes. Strikingly, the PLA analysis on brain slices at the level of the corpus callosum from animals treated with MTP-PlexinA1 in the cuprizone model showed a clear reduction (-60%) of the number of NRP1/PlexinA1 interactions compared to control animals (vehicle treated). While providing evidence that a sufficient amount of the peptide is passing the blood brain barrier and reaches the lesion sites, this result provides the demonstration of the mechanism of action of MTP-PlexinA1 to exert its protective effect by disrupting NRP1/Plexin-A1 dimerization.

3. Fig. 3C+D: Show representative data of MOG and NG2 stainings. MOG is a myelin marker and hardly stains the OL soma, making colocalization with other somatic markers challenging. Appropriate markers are e.g. ASPA or GSTPi.

The analysis of the developmental pattern of expression of Plexin-A1 has been removed. We believe that together with the immunocytochemical analysis (ex supplementary figure 2) at the electron microscope level the results can be published in a separate paper better exemplifying the timing and subtypes of cells expressing Plexin-A1, including neurons and blood vessels. The therapeutic aspect being more important after revisions this information is not crucial for the current study. It is also

simplifying the message by having one single marker of OL (CNP) in all figures dealing with the expression of Plexin-A1. We anyway provide for review only additional data confirming the expression data (see response to reviewer 1 and corresponding additional figures 3-5).

4. Antagonist: How did the authors arrive at these doses? Please provide PK/PD data. Specifically dose-response data should be included in the models relevant for the current study. i.e. cuprizone model, OliNeu cells

The doses used in vitro and in vivo were defined from previous work having shown the efficacy of this peptide in cancer settings (Jacob et al., 2016). In this study the disruption of NRP1/PlexA1 dimers was measured by proximity ligation on tumor samples after intraperitoneal injection of 1µg/kg. This suggested that this dose is indeed efficient. There are currently no technical methods for PK/PD analysis of hydrophobic transmembrane domain peptides being difficult to extract from tissues. Thus, efficient dosing has been determined empirically in different animal models based on the in vitro efficient doses. Current work in the lab and other labs are developing/optimizing suitable methods including mass spectrometry. Recent data were obtained with a very similar peptide targeting NRP1 in a QWBA study tracing the distribution of a tritium-labelled peptide. This showed that TM peptide are rapidly expelled from the blood flow to enter tissues (including the brain) in less than 5 minutes without accumulation in urines. We also obtained data showing the good plasmatic stability of MTP-PlexA1 with more than 60% integrity after two hours. Here, we point that as in the cancer setting, there is a clear reduction of NRP1/PlexA1 dimers in the brain of treated animals also suggesting that the dose is efficient. The results obtained with 10µg/kg in the EAE model suggest that the activity could be strengthened with increased doses. Our goal is to publish a full study to address these important questions of PK/PD including a formulation study. However, the effects are dose dependent both in vitro (see revised figure 6) and in vivo (see figure 12). Future investigations will focus on the definition of the best dose and administration schedule.

Minor points:

1. Page 6, Reference "Mi": Delete "1" Done
2. Page 5, Results 1st paragraph: Rephrase: e.g. white matter oligodendrocytes Done
3. Scale bars in Fig. 1 are either confusing or completely missing New scale bars have been added
4. Page 6, CNP is a marker of mature OLs. Co-localization studies with OL lineage cell markers during development should use a pan-OL marker in addition to take this into consideration. The developmental pattern will be further analyzed elsewhere. We believe CNP as a robust marker of OL and adapted to our demonstration of Plexin-A1 expression in pre-myelinating OL.
5. Page 7, "Moreover, similarly to what has been described in human samples of multiple sclerosis." - watch syntax; This is not a full sentence. Rephrase. This sentence has been edited
6. Fig. 6B, What mRNA levels are shown? This is corrected on the figure (MBP mRNA expression level)
7. Page 10, "Results obtained with forelimbs and hindlimbs (see supplementary Fig. 2) were significantly not different at the same time points," - This doesn't make sense. Rephrase: e.g. did not differ – This sentence has been edited
8. Generally: Use english decimal presentation "." instead of "," The decimal presentation has been changed
9. Page 11, "maximal phase of myelination" - Rephrase: e.g. peak myelination Done
10. Fig. 1, Clearly state that the double immunohistochemical presentation maybe possible to inform co-localization because both antigens were detected in the same section. It should be noted that though that these data are not reliable as they were obtained employing non-confocal microscopy The staining was achieved on adjacent sections. The false color process was used to identify double positive cells.

Additional Figure 1

(A) Negative control without first antibody for PlexinA1 and CNP

(B) Western blot showing mouse and human Sema3A, tagged with His-Tag and revealed with anti-Sema3A or His antibody to demonstrate specific cross reactivity

(C) Immunostaining of PlexinA1 and CNP on a normal and tumoral brain tissue array (BNC17011b US Biomax)

Additional figure 2 . Expression of Plexin-A1 and SEMA3A in multiple sclerosis patients versus healthy controls.
 Full western blots showing brain samples lysates of multiple sclerosis patients and healthy controls revealed with anti-Plexin A1 antibody.

Agilent analysis of protein used for the western blot showing degradation especially at the high molecule size.

*** Additional Figures 3, 4, & 5 for referees - not shown.**

Thank you for the submission of your revised manuscript to EMBO Molecular Medicine. We have now received the enclosed report from the referee who was asked to re-assess it. As you will see the reviewer is now supportive and I am pleased to inform you that we will be able to accept your manuscript pending the following final amendments:

Please address the minor changes commented by referee 2. Please provide a point-by-point letter INCLUDING my comments and the reviewer's reports and your detailed responses to their comments.

REFEREE REPORTS

Referee #2 (Comments on Novelty/Model System for Author):

The authors provide data using a novel antagonistic peptide for an inhibitory pathway important for regenerative aspects of affected myelination in two model of the debilitating disease MS. A preliminary PK and toxicology profile of the drug is provided and shows tolerability in vivo.

Referee #2 (Remarks for Author):

The authors have vastly improved the quality of their manuscript by adding PK data for their drug in an orthogonal MS model. They have added data where appropriate and deleted unsupportive data initially present. The authors addressed all major concerns of the reviewers in a satisfactory way. There are still some points that need addressing:

My specific comments are outlined below and refer to page numbers of the pdf:

1. Page 3 "...only few CNP-positive OL expressed Plexin-A1 in the adult." Reference to adult is confusing as this has no relevance to a developmental aspect
2. Page 4 and elsewhere: italicize the gene names according to international nomenclature when describing RNA data. This will clearly differentiate them from the protein data
3. Suppl Fig 1: Sema 3 A looks for like 2.5-fold upregulated - not 1.6-fold as indicated in the results section
4. use decimal point throughout but also in Fig 2C in healthy control cell.
5. Rearrange Panels in Fig. 5 so that the old panel A will be panel C. This would be consistent with the flow of the results.
6. Fig 6A: correct label of drug concentrations to either 1x10 or rather delete the number 1 and the decimal point
7. correct label of drug concentrations to either 1x10 or rather delete the number 1 and the decimal point
8. Page 8: "DRAD": explain acronym

Point by point response

1) Please address the minor changes commented by referee 2. Please provide a point-by-point letter INCLUDING my comments and the reviewer's reports and your detailed responses to their comments (as Word file).

All points are addressed in the final version of the manuscript and detailed herein.

REFeree REPORTS

Referee #2 (Comments on Novelty/Model System for Author):

The authors provide data using a novel antagonistic peptide for an inhibitory pathway important for regenerative aspects of affected myelination in two model of the debilitating disease MS. A preliminary PK and toxicology profile of the drug is provided and shows tolerability in vivo.

Referee #2 (Remarks for Author):

The authors have vastly improved the quality of their manuscript by adding PK data for their drug in an orthogonal MS model. They have added data where appropriate and deleted unsupportive data initially present. The authors addressed all major concerns of the reviewers in a satisfactory way. There are still some points that need addressing:

My specific comments are outlined below and refer to page numbers of the pdf:

1. Page 3 "...only few CNP-positive OL expressed Plexin-A1 in the adult."
Reference to adult is confusing as this has no relevance to a developmental aspect
This has been corrected by removing the word "adult"
2. Page 4 and elsewhere: italicize the gene names according to international nomenclature when describing RNA data. This will clearly differentiate them from the protein data
Done
3. Suppl Fig 1: Sema 3 A looks for like 2.5-fold upregulated - not 1.6-fold as indicated in the results section.
Thank you for noticing this mistake. Numbers have been corrected to 3.3 fold and 4.2 fold for the mean expression and 2.8 fold and 4.7 fold for the medians. The ancient numbers related to an earlier version of the analysis with less cases.
4. use decimal point throughout but also in Fig 2C in healthy control cell.
Done
5. Rearrange Panels in Fig. 5 so that the old panel A will be panel C. This would be consistent with the flow of the results.
Figure 5 has been combined with Figure 4. The new figure 4 is respecting the flow of the results (main text has been modified to fit with the new figure)
6. Fig 6A: correct label of drug concentrations to either 1x10 or rather delete the number 1 and the decimal point
Done
7. correct label of drug concentrations to either 1x10 or rather delete the number 1 and the decimal point
Done
8. Page 8: "DRAD": explain acronym
Done (Radial diffusion)

Corresponding Author Name: BAGNARD Dominique

Manuscript Number: EMM-2019-10378-V2